psychology/cognition

body representation, body perception, visual system, adaptation, perceptual size distortion, multisensory integration

**Author for correspondence:**
Sarah D'Amour
e-mail: sdamour44@gmail.com

# Changes in the perceived size of the body following exposure to distorted self-body images

Sarah D'Amour, Deborah Alexe and Laurence R. Harris

Centre for Vision Research, York University, Toronto, Canada

  SD, 0000-0002-9383-0577; LRH, 0000-0002-7154-8757

Inaccurate perceptions, such as under- or over-estimation of body size are often found in clinical eating disorder populations but have recently been shown also in healthy people. However, it is not yet clear how body size perception may be affected when the internal body representation is manipulated. In this study, visual adaptation was used to investigate whether exposure to distorted visual feedback alters the representation of body size and how long any such effects might last. Participants were exposed for five minutes to a distorted life-size image of themselves that was either 20% wider or 20% narrower than their normal size. Accuracy was measured using our novel psychophysical method that taps into the implicit body representation. The accuracy of the representation was assessed at 6, 12 and 18 min following exposure to adaptation. Altered visual feedback caused changes in participants' judgements of their body size: adapting to a wider body resulted in size overestimation whereas underestimations occurred after adapting to a narrower body. These distortions lasted throughout testing and did not fully return back to normal within 18 min. The results are discussed in terms of the emerging literature indicating that the internal representation of the body is dynamic and flexible.

## 1. Introduction

Perceptual biases of body size are influenced by different environmental stimuli and by experience. However, it is important that the brain accurately represents the body's dimensions because knowledge of accurate body size is essential for many activities in everyday life. Underestimation or overestimation of body size can occur due to perceptual disturbances, such as those found in patients suffering from eating disorders [1]. But perceptual disturbances of body representation are also found in healthy

people (e.g. [2–4]. Therefore, it is also crucial to understand the mechanism that underlies any such inaccuracy in the non-clinical population. Knowing how the body is represented in the brain and how the representation is correctly scaled is critical for a complete understanding of proprioception and motor functioning [5]. When referring to the body representation we consider it as a kind of memory, along with other related concepts such as the perceived 'normal' body size [6].

There are two important processes that impact body representation [1]. First, the integration of various sensory inputs such as tactile, kinesthetic and visual cues to body size. Secondly, there are non-sensory factors including cognitive factors based on prior knowledge of the body or on individual beliefs. This study investigates the former. Does visual feedback contribute to the perception of body size (for example when looking in a mirror)? Can distorting this visual feedback influence the representation of body size? We also include a biopsychosocial approach and consider sex and body dissatisfaction.

## 1.1. Visually modifying perceived body size

Visually modifying the perceived size of the body is known to impact tactile perception [7], tactile distance perception [8] (but see [9]), tactile size perception [10], haptic perception [11], pain perception [12,13], the perceived size of objects and their perceived distance from the observer [14], the rubber hand illusion [15] and motor control, such as grasping [16,17]. These observations all suggest that visually changing perceived body size can indeed alter a person's mental representation of their body and that these changes can affect perception. Distorting perceived body size and shape visually [8,11,13,17,18], proprioceptively [19–23] or with cutaneous anaesthesia [24] provides further evidence of the complex relationship between sensory perceptions and the body representation to which it is referenced. However, none of these studies has used natural-sized images of the participant themselves. Life-size images would seem to be the easiest to identify with as, of course, that is the size one sees when looking into a mirror—the usual way of seeing oneself. Therefore, we set out to use visual adaptation using life-size natural images of the participants themselves.

## 1.2. Visual adaptation

There has been extensive research looking at the effects of adaptation on sensation and perception. Even within the realm of body perception, research is wide ranging and varied (see review [25]). For example, Ambroziak et al. [26] and Sekunova et al. [27] used avatar images presented on a small computer screen and Stephen et al. [28–30] used small photographs with the face hidden by a black square. Most studies adapted their participants with photographs, avatars or drawing of other people and investigate the influence of various cultural factors such as race [31], media and society [32,33], pose [27] or gender [34] on body perception. These and other studies have shown that visual adaptation can alter the visual perception of a participant's own body [35]. However, the low-level effects of visual adaptation are generally a brief change that occurs in perception or sensitivity following exposure to a specific stimulus [36]. Even a brief exposure to a distorted image of a body (e.g., to a fat or thin silhoutte) has an impact on perceived body size and can subsequently cause individuals to over- or underestimate their own size [35,37,38]. Generally, after participants view a thinner picture of themselves, it leads them to judge their perceived body size to also be thinner, although the polarity of the effects reported is not always clear. However, the time course of the effects of visual adaptation on body size have not been reported. Cazzato et al. [39] is perhaps the closest, but they showed only that adaptation to a round body caused an increase in the liking of round figures for about 10 min. One of our aims was to measure the time course for the first time.

We used the novel psychophysical method described in our previous studies [2–4]. In this method, we show two sequential images—a reference image and a test image—and ask the participant to judge which image most closely resembles their idea of their own body. The width of the test image is adjusted until both the test and reference images are chosen equally—critically, neither of these images matches their actual idea of themselves. The width of the participant's body representation is then taken as the width halfway between the widths shown in the two photographs. Participants might transiently see an image size that actually matches their own idea of their size, but the image size would then be rapidly adjusted away from this value towards the image size that was equally like their idea of their own size as the reference image. Thus, this method gives us a best estimate of their body representation, as ultimately in the procedure, they are forced to rely on accessing their internal body representation to decide which of two photographs, neither of which actually matches this representation, is closest to it. Using this psychophysical procedure, participants' body representation was first assessed to obtain baseline

accuracy. They were then exposed to 5 min of a distorted image of their own life-sized body that was either 20% wider or 20% narrower than their actual size. We expected that altering their visual feedback in this way would cause changes in the size of the body's representation, with biases appearing in the direction of the adapting stimulus—adapting to a wider body would result in body size representation overestimation (thinking they were wider than they were) whereas underestimations (thinking they were narrower than they were) would occur after adapting to a narrower body. In order to see if sex or body satisfaction made people more or less vulnerable to adaptation effects, we tested both men and women and assessed their body satisfaction levels. We also measured the time course of any adaptation effects. The size of the body representation was assessed at 6, 12 and 18 min following adaptation. We also expected that there would be differences between the adaptation directions, such that adapting to a wider body would cause greater changes than adapting to a narrower body. This was expected because of the preponderance of clinical cases [40] and normal studies [41] in which people can tend to feel fatter than they are. and the relative rarity of the opposite effect. It was important to measure the time course to contrast any changes to the internal representation with low-level adaptation effects that typically last only a few seconds after exposure.

# 2. Methods

## 2.1. Participants

Thirty participants took part in the experiment (17 females, mean age 24.1 years, s.d. = 7.91 years). They were recruited from the York University Undergraduate Research Participant Pool and received course credit for taking part in the study. The experiment was approved by the York Ethics Board and all participants signed informed consent forms. The study was performed in accordance with the Declaration of Helsinki (2003).

## 2.2. Materials/stimuli

### 2.2.1. Body dissatisfaction

We measured body dissatisfaction using the Body Shape Questionnaire (BSQ) [42]: a 34-item self-report questionnaire that was developed to assess concerns about body shape and experiences of feeling fat that participants may have experienced within the previous month. Each question uses a scale from 1 (never) to 6 (always) that are added together to obtain a total BSQ score ranging from 34 to 204, with lower scores indicating lower levels of body dissatisfaction and higher scores indicating higher levels of body dissatisfaction. The test was administered before the experiment began to obtain a measure of body dissatisfaction. The total BSQ score was used for correlations. To create categorical groups, participants were divided into high and low groups according to whether their score was above or below the overall median.

### 2.2.2. Photographs

Colour photographs of each participant's whole body were taken using a digital camera (Canon EOS Rebel T6; flash on; no zoom function), with a camera distance of 270 cm. Participants were asked to stand in front of a white wall, wearing a standardized black outfit (figures 1 and 2) that was provided to obtain accurate outlines of their size and shape. The images were then corrected for any lens distortions, cropped to include only the whole body and formatted on a white background. The image served as the undistorted reference image and was used for creating the distorted images. Actual body height was measured from the bottom of the feet to the top of the head using a ruler taped up to a wall. The image was presented life-size, projected (using a BenQ 1080p short-throw projector) onto a screen at a viewing distance of 270 cm by digitally adjusting the magnification of the image until it physically matched the participant's actual body size. The viewing distance was chosen as matching the camera's focal length multiplied by the magnification [43] which minimizes distortions.

### 2.2.3. Distorting the images

Images were presented and distorted using Matlab (v. 2011b) and PsychToolbox [44] running on a MacBook Pro. Adaptation stimuli were created by altering the width of the image making it 20%

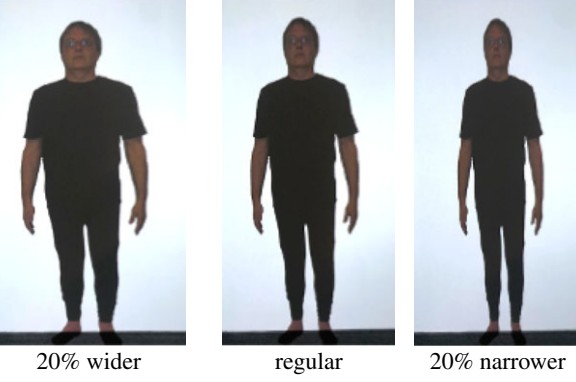

| 20% wider | regular | 20% narrower |

**Figure 1.** Example of altered visual feedback stimuli. Participants were exposed for 5 min to a distorted image of their body that was either 20% wider (left) or 20% narrower (right) than the undistorted image (centre). The images are of one of the authors.

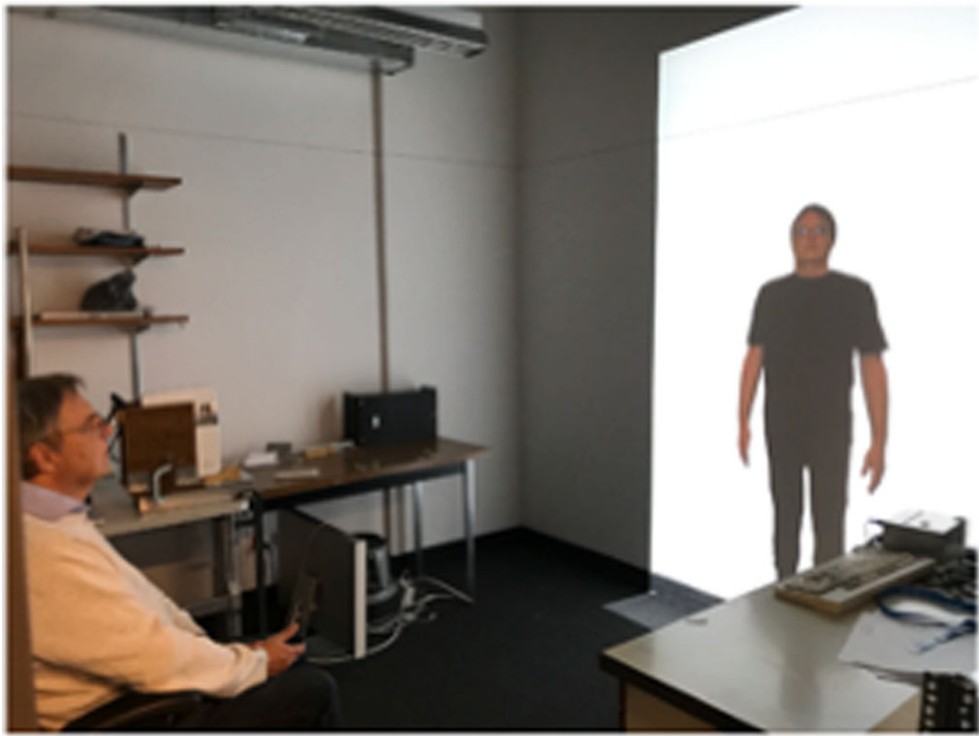

**Figure 2.** Visual adaptation experimental set-up. Participants viewed a full-size image of themselves arranged to be standing natural on the floor. The image is of one of the authors.

wider or 20% narrower than the original as shown in figure 1. During the assessment phase the width of the test image was distorted (made either wider or narrower) using a QUEST adaptive staircase psychometric procedure [45]. The test image started either 20% wider or narrower than the reference image with a standard deviation of the QUEST's guesses set to 30%. The images were viewed in the centre of a projector screen with the full body shown from the front view arranged to be standing on the floor (figure 2). The height of the photograph was set to the participant's actual height.

## 2.3. Procedure

To assess the baseline size of a participant's body representation, they sat in a chair approximately 270 cm from the projector screen (figure 2). Each trial consisted of two 1.5 s intervals—one interval containing an undistorted natural-size image of themselves (the reference image) and one interval containing a distorted image. The presentation of the two images was separated by a blank white screen presented for 1.5 s. Participants identified which interval contained the image that most closely

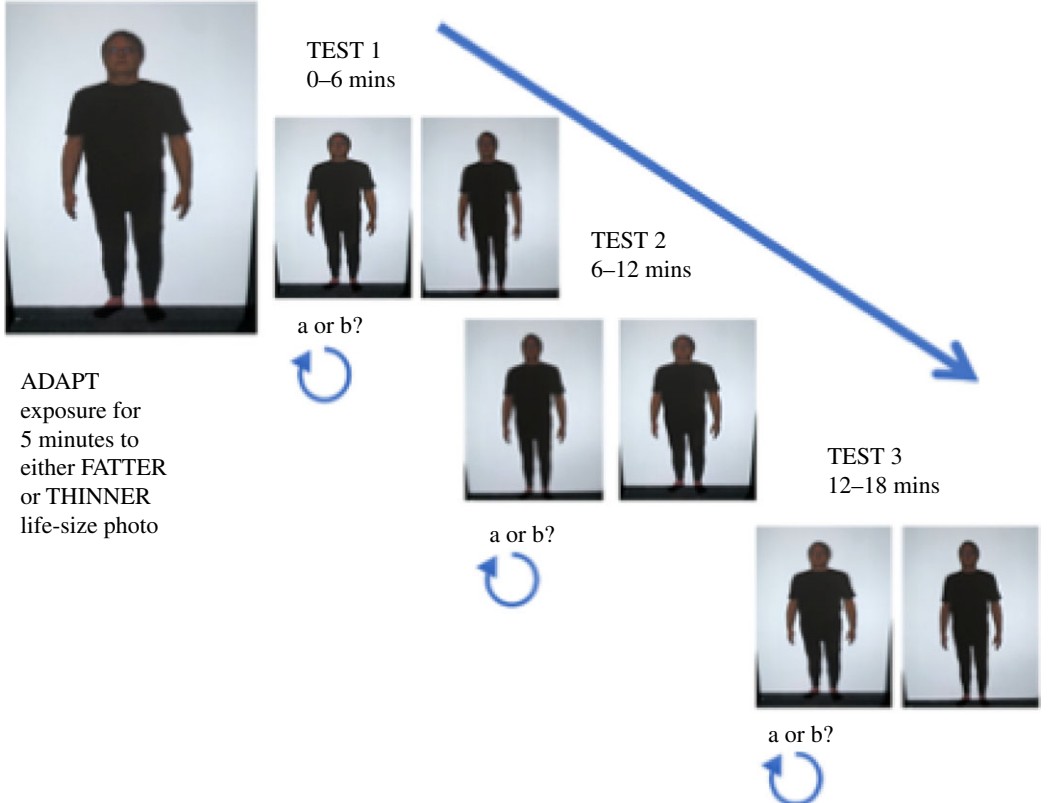

**Figure 3.** Visual adaptation experimental design. Following the five minutes of adaptation, perceived size accuracy was assessed three times to obtain measures at 6, 12 and 18 min to get a time course for any adaptation effects. The images are of one of the authors.

matched their perception of their own body and responded using a two-button computer mouse (left button for first interval, right button for second interval). A QUEST adaptive staircase procedure [45] was used with a two-alternative forced choice (2AFC) design to vary the width of the distorted image. If the test image were chosen as the preferred image, the next test image in that staircase was shifted further in the direction away from the reference image (i.e. wider if it were already wider). If the reference image were chosen, the test image was altered in the direction towards the reference size. The size of the test image could thus cross from wider than the reference image to narrower. Two randomly interleaved staircases (25 trials per staircase) were used for each condition (50 trials total), with one starting with the test image 20% wider than the reference and the other with the test image starting 20% narrower. This method honed in on the width of the test image which was equally likely to be chosen as matching their perception of their own body as was the reference image.

After baseline body size perception had been assessed they were then shown a distorted image of themselves (either 20% wider or 20% narrower but at full height, figure 1) for 5 min and asked to scan around the picture. After this, the above procedure was repeated three times in succession (figure 3 shows the timeline). As each procedure took about 6 min, this allowed us to assess perceived body size at 6, 12 and 18 min after exposure to the adapting stimulus.

## 2.4. Data analysis

The QUEST program returned an estimate of the distortion needed for the participant to report that the distorted image was equally like their perceived body size as the undistorted, reference image. The algorithm assumes the observer's psychometric function follows a Weibull distribution and adaptively determines the amount of distortion to be presented based on the participant's response to the previous trials. As the experiment goes on, knowledge on the observer's psychometric accumulates. To visualize and confirm the QUEST's performance, the results of the participant's decision (0 for when test image was made wider, 1 for when the test image was made narrower) were plotted against the distortion used for each trial and fitted with a logistic (equation (2.1) and figure 4)

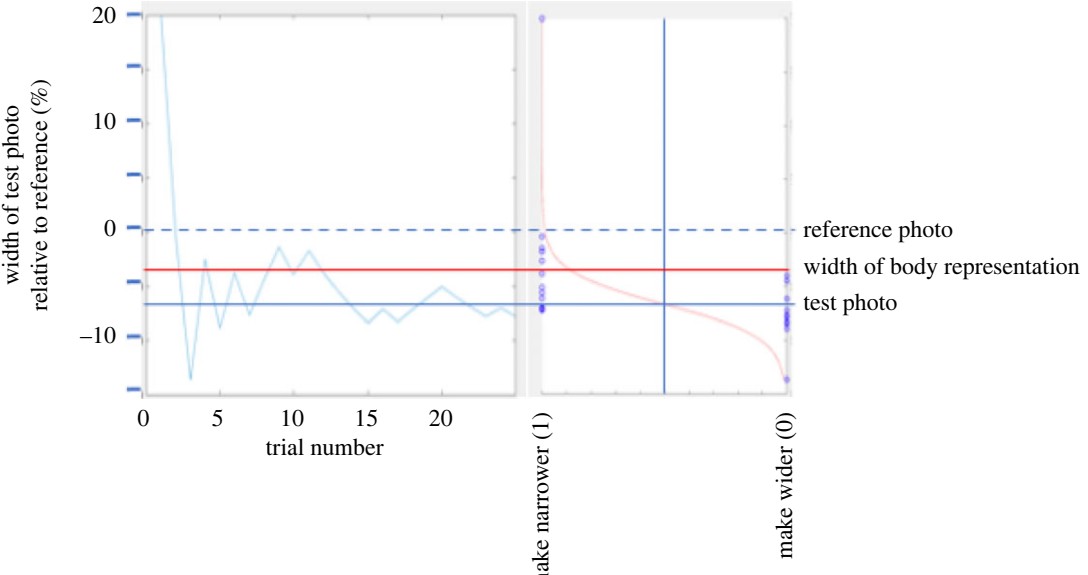

**Figure 4.** Illustrates the analysis method. On the left is a typical participant's staircase controlled by the QUEST. The dashed horizontal line shows the width of the reference photograph. On the right is the logistic function fitted to the data, using 1 when the participant's selection resulted in the test image being made narrower and 0 when the selection resulted in the test image being made wider. The vertical axis plots the width of the test photo as a percentage of the width of the reference photo. The vertical scale is the same for the two parts of the figure. The width of the body representation is taken as halfway between the reference width and the point of subjective equality obtained from the psychometric function.

using the curve fitting toolbox in Matlab.

$$\text{Decision} = 1/(1 + \exp(-(x - x_0)/b)), \tag{2.1}$$

where $x_0$ is the undistorted size, $x$ is the distorted size that was equally likely to be judged as matching the observer's body size, and $b$ is an estimate of the slope of the function. The size of the internal body representation was taken as the point halfway between $x$ and $x_0$ which were each judged as equally like their internal representation. We then subtracted $x_0$ from this value to derive a difference-from-accurate score where positive numbers corresponded to an overestimate and negative numbers to an underestimate. Figure 4 shows a typical staircase and the corresponding logistic fit. The values used in the statistical analysis were obtained by pooling the responses from two staircases (25 points each).

All of the 30 participants' data for each condition were examined to validate the accuracy and efficiency of the QUEST procedure. The QUEST's performance was also checked to determine whether reliable estimates were obtained within 50 trials. All data were inspected for outliers. Outliers were defined as points that fell outside ±3 standard deviations from the mean. All the data from a participant that failed to meet these criteria were discarded from the analyses (one participant—one male).

Baseline accuracy values from the control condition were subtracted from the adaptation values for each condition to obtain 'change score' values that were used for data analyses. One-sample $t$ tests were conducted for each condition to assess whether difference-from-accurate values significantly differed from zero (accurate). Mixed measures analyses of variances (ANOVAs) were used for statistical analyses, with α set at $p < 0.05$ and *post hoc* multiple comparisons were made using Bonferroni corrections.

## 3. Results

The mean baseline accuracy of our participants' estimates of their body width was $0.40\% \pm 1.54\%$. Figure 5a shows the results for perceived body width relative to their individual baselines after adaptation to a wider and narrower body. One-sample $t$ tests were conducted to see if the adaptation values were different from baseline at the different time points. Adapt wide was significantly different from their baseline scores for all three test sessions (all $p$s < 0.05). Adapt narrow was significantly different from baseline for the first test session ($p = 0.001$), trending for the second ($p = 0.082$), and

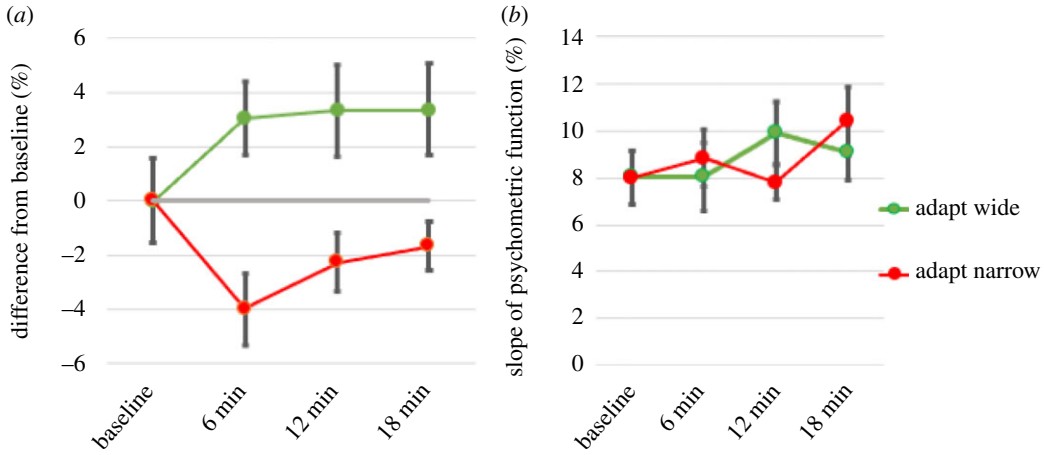

**Figure 5.** Visual adaptation results. (*a*) shows the percentage changes relative to each person's control levels measured immediately before adaptation. (*b*) shows the slopes of the psychometric functions (*b*s in equation (2.1)). Green lines: after adapting to wide stimulus, red lines: after adapting to narrow stimulus. Error bars are s.e.s.

showed no difference by the third ($p = 0.128$). Figure 5*b* shows the slopes (see Methods) of the psychometric functions. There were no significant differences from baseline for any time or direction.

A four-way mixed ANOVA was conducted to test for the within-subjects effects of altered visual feedback type (wider or narrower), time (6, 12 or 18 min) and the between-subjects effects of sex (male and female) and BSQ group (low and high). There was a significant effect for type (wider/narrower), $F_{1,25} = 17.47$, $p < 0.001$, $\eta_\rho^2 = 0.411$, and for time, $F_{2,50} = 3.90$, $p = 0.027$, $\eta_\rho^2 = 0.135$. A significant interaction was also observed between type and time, $F_{2,50} = 5.02$, $p = 0.011$, $\eta_\rho^2 = 0.167$. There was a significant four-way interaction between type, time, sex and BSQ group, $F_{2,50} = 4.15$, $p = 0.022$, $\eta_\rho^2 = 0.142$. *Post hoc* follow-up statistics to break down the four-way interaction showed that in the low BSQ group, females differed during times 1 and 2 when adapted small (−5.89% ± 2.28, $p = 0.048$, 95% CI = [0.05, 11.73]). Type differed in the low BSQ group for females only at time 1 (9.91% ± 3.64, $p = 0.012$, 95% CI = [2.41, 17.40] and for males at time 1 (7.39% ± 3.15, $p = 0.027$, 95% CI = [0.90, 13.88] and time 2 (6.45% ± 2.82, $p = 0.031$, 95% CI = [0.64, 12.26]. Differences in type were also found in the high BSQ group for females at time 2 (6.21% ± 2.41, $p = 0.016$, 95% CI = [1.25, 11.16] and time 3 (7.47% ± 2.36, $p = 0.004$, 95% CI = [2.61, 12.34] and for males only at time 1 (11.24% ± 4.46, $p = 0.018$, 95% CI = [2.06, 20.41]. Figure 6 shows the interaction broken down by the factor levels and groups.

## 4. Discussion

This research investigated how exposure to distorted visual images of their own natural-sized body impacted the participant's perception of body size. After adapting to a distorted wider image of their body, people's perception of their bodies became wider. After adapting to a distorted narrow image of their body, participants thought that they were narrower than they actually were. These distorted perceptions of themselves did not return back to normal by the end of the testing sessions (18 min).

We interpret these effects as resulting from alterations in body representation as opposed to being a simple low-level perceptual effect such as when the bars of a grating appear wider after viewing a high spatial frequency (narrowly spaced) grating. Firstly, such low-level effects are often short lived [36]. Secondly, such adaptation generally produces a negative after-effect in which subsequent visual stimuli are perceived as shifted in the opposite direction to the adapting stimulus. Though we found differences between the variables and groups we tested, everyone experienced changes in perceived size after adaptation in the positive direction (the same direction as the viewed distortion). If our effects were caused by everything appearing, for example, narrower (and therefore needing to be made larger to compensate), then such a global effect would also have shown in the slopes of the psychometric functions (adapt wider, slopes shallower; adapt narrow, slopes steeper) as the functions were stretched or compressed. Figure 5*b* shows that this was not the case. Thirdly, we found varying effects related to the different factors and groups considered in this study. If our adaptation effects were due only to low-level visual perception adaptation mechanisms, then they should depend only on the observer's viewing history and not on individual factors such as their level of body

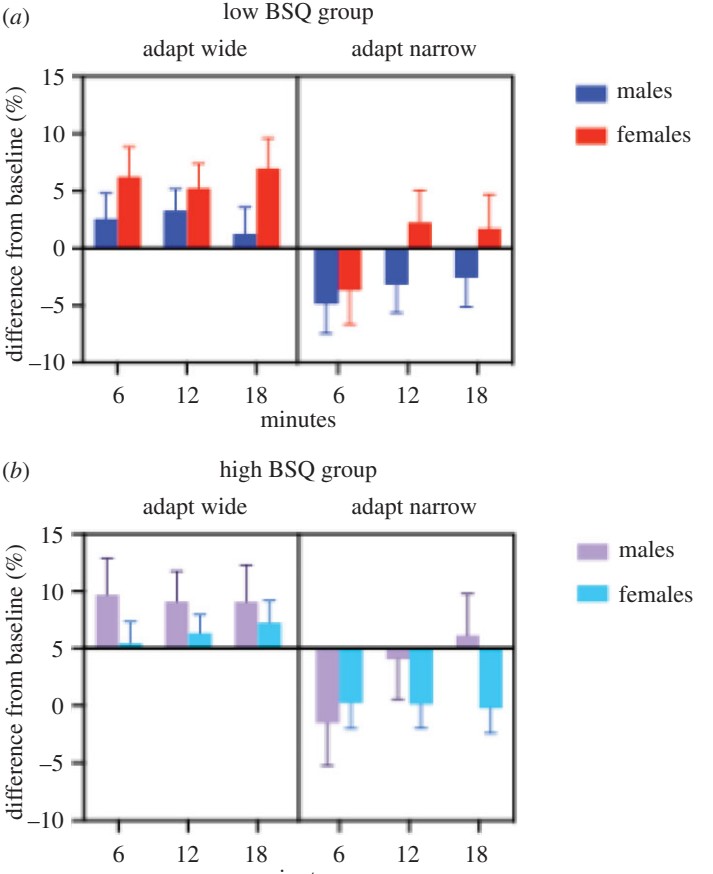

**Figure 6.** Visual adaptation results broken down by all factors. The graph shows the effect of adaptation after adapting wide (left) and narrow (right) for males (dark blue and purple) and females (red and pale blue) for the low BSQ group (*a*) and high BSQ group (*b*).

satisfaction. Instead, we speculate that after viewing themselves as, for example, wider, this distorted dimension becomes incorporated into their brain's representation of their own body. When asked subsequently which image most closely matches their own idea of their body, they therefore select, in this example, a wider image as being equally like their body representation as the reference image.

We interpret our results as a positive effect on the body's representation (viewing a wider image causing the internal representation to become wider) but appreciate that this could be a consequence of the photographs themselves becoming perceptually distorted [6,26]. That is, after exposure to a wider body, the test photographs could have appeared narrower, meaning a wider one would need to have been chosen to match their own body representation. Our technique of having our participants viewing images that did not actually match their perception of their own body makes this less likely—and experiments should be repeated with reference images even more dissimilar to their expected settings. We would also expect such perceptual distortion of the photographs to show as changes in the slopes of the psychometric functions which were not seen (figure 5*b*). Again, we call on the variation in our results with high-level non-sensory items (body satisfaction and sex) to suggest that perceived image distortion cannot on its own explain our results.

Although distortions were found after adaptation independent of sex or body dissatisfaction, there were differences between the groups (figure 6). Surprisingly perhaps, greater distortions occurred in general for males, although large effects were also seen for females in the low BSQ group when adapting wider, and for females in the high BSQ group when adapting narrower. It has been shown that the effect of adaptation to fat body images is smaller for females with higher body dissatisfaction compared with those with less body dissatisfaction [37,46]. This could be related to the distortions commonly found in clinical disorders [47,48] or perhaps to the differences in average body size between males and females [49–51].

A recent review by Brooks *et al.* [52] reviewed several key points surrounding adaptation in body size and provided support for high-level adaptation of body representation. For example, Hummel *et al.* [38]

showed that adaptation after-effects to wide bodies affected the subsequent perception of a participant's own body but adaptation to wide and narrow black rectangles caused no such perceptual distortion for subsequently viewed bodies. Such observations support the idea of adaptation to images specifically of distorted bodies causing their effects at a high level in the brain in areas specific to body shape, and not just a low-level effect of width in general. Media exposure to idealized bodies has also been shown to affect body size estimation through high-level alterations in response to real-world experiences [6,9]. Furthermore, adaptation effects are maintained despite changes in body orientation [53]. Thus, for these reasons and with the support of studies such as those reviewed by Brooks *et al.* [52], we conclude that our adaptation effects were probably the results of changes at the level of the body representation in the brain, although other high-level explanations are possible, including real-world experiences [6,25,37,53], concepts of attractiveness [54], changes in body dissatisfaction or other perceptual biases, such as have been shown for faces [55].

# 5. Conclusion

We have shown that visual feedback impacts the internal representation of body size, and that visual input and exposure is involved in how we generate our perception and representation of our bodies. Our study fits into the emerging literature showing that body representation is flexible and can be changed dynamically.

Ethics. This study was carried out in accordance with the recommendations of the York Ethics Board. The protocol was approved by the York Ethics Board (certificate e2017-196). All subjects gave written informed consent in accordance with the Declaration of Helsinki (2003).

Data accessibility. The datasets supporting this article have been uploaded as part of the electronic supplementary material [56].

Authors' contributions. S.D.: conceptualization, data curation, formal analysis, investigation, methodology, project administration, software, supervision, validation, visualization, writing—original draft, writing—review and editing; D.A.: data curation, investigation, writing—review and editing; L.R.H.: conceptualization, data curation, formal analysis, funding acquisition, methodology, project administration, resources, software, supervision, validation, visualization, writing—original draft, writing—review and editing.

All authors gave final approval for publication and agreed to be held accountable for the work performed therein.

Conflict of interest declaration. We declare we have no competing interests.

Funding. This work was supported by the Natural Sciences and Engineering Council of Canada (NSERC), grant no. 46271-2015.

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
