## [Peer Review File · Royal Society Open Science]

Review History

RSOS-210722.R0 (Original submission)

Review form: Reviewer 1

Is the manuscript scientifically sound in its present form?

Yes

Are the interpretations and conclusions justified by the results?

No

Is the language acceptable?

Yes

Do you have any ethical concerns with this paper?

No

Have you any concerns about statistical analyses in this paper?

No

Recommendation?

Major revision is needed (please make suggestions in comments)

Comments to the Author(s)

The general topic of the paper is really interesting: how viewing images of oneself (e.g. in the mirror) influences our perception of body size. The authors review past research indicating that visual feedback about body size can affect body representation, as well as studies showing that adaptation to images of bodies can cause (attractive) perceptual aftereffects when making judgements about one's own body size. The current study seems to differ mainly in showing participants life-sized images of their own body. Adaptation to images of oneself with a distorted horizontal width impacted subsequent judgments about the 'correct' width of these images (and these were attractive aftereffects). I didn't get much of a sense from the paper about what mechanism might underlie these effects. I found some ideas in the paper to be a bit under-explained. Specific comments below.

Introduction:

- An idea conveyed in the Introduction is that the psychophysical task that the experiment uses taps into higher-level body representation rather than lower-level visual adaptation. However, the design of the task seems like it could capture lower-level visual aftereffects perfectly well. i.e., you are adapted to a stretched or compressed image of a body, then you are tested on similar images presented in the same area of the visual field. In the Discussion, there are some arguments for why the results might reflect adaptation at a higher level, but it doesn't seem to be the case that the task itself is designed to isolate higher-level from lower-level effects, or to isolate effects specific to body representation from adaptation to object size, shape or aspect ratio more generally.

- p6 The term "body image" is used in contrast to "body representation", but it's not explained what the difference is.

- p7. "We also expected that there would be differences between the adaptation directions - such that adapting to a wider body would cause greater changes in accuracy than adapting to a narrower body". Why did you expect that? Similarly, the reason for testing the timecourse of the effect (testing at 6/12/18 minutes) isn't explained.

Results:

- I don't see any results reported about how accurate perception of body size was at baseline and how much this varied between participants. Similarly, I was a bit confused by the first paragraph of the results, where it's stated that 1-sample t-tests were conducted to test differences in perceived body size from 'accurate' (which I read to mean actual body size), but then the tests actually contrast performance following adaptation to perceived size in the baseline condition, which presumably might not be accurate.

- I wonder if the authors could comment on the size of the effects reported. For example, by expressing the apparent effect of adaptation on perceived image size in terms of change in the size of the body (e.g. in centimetres or waist size) or visual angle. Also, in Figure 4, the y-axis label is perhaps ambiguous in whether the scores are expressed as a percentage of the veridical image size or a percentage of the image size that the staircase converged on at baseline.

- Why were males/females and BSQ group included as factors in the ANOVA? What difference did you expect to see between males and females? There was a significant 4-way interaction in the ANOVA, but I could have used some more text here to highlight what is driving that

interaction, or what I should be looking for in the complex plot of the data broken down across all the conditions.

Discussion:

- “We interpret these effects as altering the body representation as opposed to being a simple perceptual effect that resulted from fatiguing particular units or channels”. It would be good to define a little more specifically what is meant here. The argument seems to be that the effect is not specifically visual adaptation (or lower-level visual adaptation as opposed to higher-level visual adaptation?), but relates to more abstract/high-level/multisensory representations of the body. Some of the arguments here are not convincing to me. First, visual adaptation effects can be long-lasting (e.g., one study that comes to mind reports gaze aftereffects, which are commonly interpreted in terms of changes in the sensitivity of sensory channels, lasting up to 24 hours, Kloth & Rhodes, 2016, JEP:HPP), so the 18-minute duration doesn’t seem like an argument against the effects being visual or relying on changes in the sensitivity of sensory channels. Second, if you want to argue that the effects being attractive rather than repulsive is significant, it would be good to say something about what the underlying mechanism of an attractive aftereffect might be. If it’s not ‘fatigue’ of sensory channels causing the effect of adaptation, then what else might it be? Third, it seems like the kind of approach of the cross-adaptation studies that you refer to later on (adapting to bodies vs rectangles) would be a way to test this question of how high vs low-level the effects are. e.g., if adapting to a distorted image of one’s own body also affected the perceived width of non-body objects, then this might suggest it is a lower-level visual effect.

- The manipulation is the horizontal stretch of the image (with the aspect ratio changing). So in the larger adapter image, the face features are stretched unnaturally in the horizontal dimension, as well as the rest of the body. There is a literature on adaptation to faces distorted in this kind of way (e.g., some of Mike Webster’s work; ‘figural aftereffects in face perception’). There also appears to be research on adaptation to the aspect ratio of objects more generally (Storrs & Arnold, 2017; <https://doi.org/10.1037/xhp0000292>). I wonder if you want to connect your work to this, as it seems closely related to the image manipulation you are using. Could your results be explained by adaptation to the aspect ratio of a familiar object? Or adaptation to the configuration of face features?

Review form: Reviewer 2

Is the manuscript scientifically sound in its present form?

No

Are the interpretations and conclusions justified by the results?

No

Is the language acceptable?

Yes

Do you have any ethical concerns with this paper?

No

Have you any concerns about statistical analyses in this paper?

No

Recommendation?

Major revision is needed (please make suggestions in comments)

Comments to the Author(s)

This manuscript describes an experiment showing the effect on body representations caused by exposure to full-height body stimuli depicting the participants own body digitally widened or narrowed by 20%. Participants perform a QUEST-directed 2AFC, choosing which of 2 test stimuli appear most accurate, and curve fitting establishes the width of the stimulus that matches participants' perception of their own body. While this experiment is interesting and topical, connecting with a recent surge of articles on this subject, there are several problems that need to be addressed.

MAJOR ISSUES

The first concerns definitions of the essential concepts for this study. While the title refers to "the internal representation of the body," and the abstract mentions "the implicit representation of the body", the meaning of these phrases is never defined. On pg 6, line 18, the authors make it clear that it's not the same as the body image (which is also not defined, even though this phrase has been used to mean various different things in the literature – please fix this), but don't explain in what way the two are different. It doesn't seem to refer to the current perceptual "readout" either (e.g. the "online" representation of the stimulus being viewed, such as the adaptation body or the test stimulus). (Please also explain what is meant by "central effects" – pg 6 line 14.). Without these definitions, it is difficult to fully comprehend what the thrust of the paper is, but it seems that it may be similar to recent work by Ambroziak et al., who distinguished the possible effect of adaptation on the perception of the test image vs. the effects on "internal body image". Clearly the authors have read this study, yet it is only given a token citation, saying "Ambroziak et al. (2019) used avatar images presented on a small computer screen". Their approach and findings are not covered. This is surprising, given that the two papers seem to address at least similar issues.

Another recent study to discuss the effects of adaptation on different body representations is Brooks et al. (2021). This article also relates to the description of the pattern of results, which is at odds with most adaptation papers. In the current study, the authors write: "Generally, after participants view a thinner picture of themselves, it leads them to judge their perceived body size to also be thinner...". In fact, the opposite is true (see Hummel et al, 2012; Brooks et al., 2016 already cited, plus Bould et al., 2018 & 2020 not yet cited). The description of the results as overestimation or underestimation is also discussed in Brooks et al (2021) and relates directly to the issue of what kind of representation is being affected by the process of adaptation. This problem affects many parts of the manuscript, including the description of results and discussion, e.g. on pg 17 lines 16-22. The results of previous body adaptation experiments show repulsion as described here (and as in low-level adaptation experiments, plus face adaptation results), and I believe that the current results may be in line with these (although given the problems with the methods, I can't be sure). If the results of the current study are out of step with previous body adaptation studies, then an explanation for this discrepancy needs to be given.

The second major issue concerns the methodology, the details and rationale of which needs to be explained in full.

There is insufficient detail on the procedure and data analysis. For example, we are told "Participants identified which interval contained the image that most closely matched their perception of their own body", yet the Data Analysis section states: "To visualize and confirm the QUESTs performance, the participant's decision (0 for smaller, 1 for bigger) was plotted against the distortion used for each trial...". It is not clear how responses of "distorted image"/"undistorted image" are transformed into "smaller"/"larger". Further, it is not clear why a logistic function is used for curve-fitting when QUEST's underlying assumption is that

performance reflects an underlying Weibull function. In any case, an example psychometric function fit might help readers to better understand this stage of the experiment.

Further, the rationale of the method is unclear. The authors state: "In order to assess this, we used the novel psychophysical method described in our previous studies (D'Amour & Harris, 2017, 2019, 2020). This method, since it does not involve participants ever actually seeing the image-size that they judge as their own, gives us a direct estimate of their body representation as opposed to their body image which is vulnerable to many cognitive factors." Given that the QUEST procedure involves the presentation of many different sized bodies, it seems highly likely that each participant will see the image size that they judge as their own on several occasions. Even if they did not see a body stimulus with this width, why would this mean that they're judging the "body representation" (again - this needs to be very clearly defined)? The authors need to spell this out.

Ambroziak et al. (2019) are clear that adaptation affects perception of the test images, not the stored body image. If this is true, then in the current study, adaptation should affect the perception of both of the test images approximately equally, and hence no effect should be shown. For example, after adapting to widened bodies, both test images should look thinner than they really are. This would predict that there should be no measured effects of adaptation, unless the adaptation affects the appearance of one image more than the other. It makes me wonder what the reported effects really represent.

It is also interesting that this method is described as "novel" when it appears very similar to the method used by Sekunova et al. (2013) but I can't be certain due to the lack of details. This paper should be cited.

I would also recommend greater consistency of phrasing throughout the manuscript.

Occasionally, adaptation is referred to instead as "exposure" and sometimes as "feedback". The latter example is particularly confusing, as it does not seem to be feedback in the conventional sense, unless the participant is being (deceptively) told "this is your actual body size", which I don't believe they are (please confirm).

Adaptors should either be described as bigger/smaller (as in figures) or wider/narrower, but not both.

MINOR ISSUES

Pg 3, line 21 onwards. See also Zopf et al. 2021, who showed that visual adaptation caused an aftereffect of body size but did not affect perceived tactile distance.

Pg 6 line 1. "factor" should be "factors"

Pg 6 Line 10. There was no adaptation in the Urdapilleta et al., 2010 study.

Pg 5 line 12 "How much of this is a simple visual after effect, comparable to low-level adaptation..." This statement is strange, given that pg 18 cover the evidence that body adaptation is high level. Also, see Pg 17, line 23

Pg 6, line 7 "We also expected that there would be differences between the adaptation directions - such that adapting to a wider body would cause greater changes in accuracy than adapting to a narrower body." Why? This hypothesis needs to be justified.

Pg9 line 9 2.2.4. Altering visual feedback with images - This section seems to be procedure, not Materials/Stimuli

Pg 9 Figure 2. Here, the height of the images seems to vary between the 3 images. Please explain.

Pg 11, line 5. "separated by a blank white screen." For how long?

Pg 15 Figure 4. Remove title from graph. Legend should match text. Don't plot a point at zero,

Pg 16 line 8. These findings clearly answer the question of whether visually manipulating body representation can alter accuracy values..." I would disagree with this. The idea that representation has been manipulated is inferred, given the effects measured. It is not a given, whose effects are then assessed.

Pg 16 line 13" We interpret these effects as altering the body representation as opposed to being a simple perceptual effect that resulted from fatiguing particular units or channels." This comment seems outdated. It has been a very long time since the theory that adaptation is due to fatigue was popular or considered credible. More recent accounts suggest recalibration. A reference more recent than Blakemore 1970 should probably be given.

Pg 16, line 23 "If our adaptation effects were due only to low-level visual perception adaptation mechanisms then everyone should be similar and show comparable levels of adaptation." Again, it is clear that body adaptation is high level. But even so, the authors need to justify this statement. Why should "everyone be similar"?

Pg 18 line 7. "The body representation's susceptibility to change after such brief experience suggests body representation may be even more plastic than previously thought." Given that there have been approximately 20 studies to date that demonstrate body size adaptation, should we really be surprised?

Bould, H., Carnegie, R., Allward, H., Bacon, E., Lambe, E., Sapstead, M., ... & Munafò, M. R. (2018). Effects of exposure to bodies of different sizes on perception of and satisfaction with own body size: two randomized studies. *Royal Society open science*, 5(5), 171387.

Bould, H., Noonan, K., Penton-Voak, I., Skinner, A., Munafò, M. R., Park, R. J., ... & Harmer, C. J. (2020). Does repeatedly viewing overweight versus underweight images change perception of and satisfaction with own body size? *Royal Society Open Science*, 7(4), 190704.

Brooks, K. R., Stevenson, R. J., & Stephen, I. D. (2021). Over or Under? Mental Representations and the Paradox of Body Size Estimation. *Frontiers in Psychology*, 12.

Sekunova, A., Black, M., Parkinson, L., & Barton, J. J. (2013). Viewpoint and pose in body-form adaptation. *Perception*, 42(2), 176-186.

Zopf, R., Kosourkhina, V., Brooks, K. R., Polito, V. & Stephen, I. D. (2021). Visual body size adaptation and estimation of tactile distance. *British Journal of Psychology*. doi: 10.1111/bjop.12514

Decision letter (RSOS-210722.R0)

Dear Dr D'Amour

The Editors assigned to your paper RSOS-210722 "Changes in the internal representation of the body following exposure to distorted self-body images" have now received comments from reviewers and would like you to revise the paper in accordance with the reviewer comments and any comments from the Editors. Please note this decision does not guarantee eventual acceptance.

Please submit your revised manuscript and required files (see below) no later than 21 days from today's (ie 07-Sep-2021) date. Note: the ScholarOne system will 'lock' if submission of the revision is attempted 21 or more days after the deadline. If you do not think you will be able to meet this deadline please contact the editorial office immediately.

on behalf of Dr Isabelle Mareschal (Associate Editor) and Essi Viding (Subject Editor)
openscience@royalsociety.org

Associate Editor Comments to Author (Dr Isabelle Mareschal):

Associate Editor: 1

Comments to the Author:

Both reviewers raise issues around the theoretical rationale of the experiments (which could be made clearer by also clearly defining concepts and linking these to the relevant literature). The methodology also needs more clarification to help interpret the results. Please provide a point by point answer to all questions raised.

Reviewer comments to Author:

Reviewer: 1

Comments to the Author(s)

The general topic of the paper is really interesting: how viewing images of oneself (e.g. in the mirror) influences our perception of body size. The authors review past research indicating that visual feedback about body size can affect body representation, as well as studies showing that adaptation to images of bodies can cause (attractive) perceptual aftereffects when making judgements about one's own body size. The current study seems to differ mainly in showing participants life-sized images of their own body. Adaptation to images of oneself with a distorted horizontal width impacted subsequent judgments about the 'correct' width of these images (and these were attractive aftereffects). I didn't get much of a sense from the paper about what mechanism might underlie these effects. I found some ideas in the paper to be a bit under-explained. Specific comments below.

Introduction:

- An idea conveyed in the Introduction is that the psychophysical task that the experiment uses taps into higher-level body representation rather than lower-level visual adaptation. However, the design of the task seems like it could capture lower-level visual aftereffects perfectly well. i.e., you are adapted to a stretched or compressed image of a body, then you are tested on similar images presented in the same area of the visual field. In the Discussion, there are some arguments

for why the results might reflect adaptation at a higher level, but it doesn't seem to be the case that the task itself is designed to isolate higher-level from lower-level effects, or to isolate effects specific to body representation from adaptation to object size, shape or aspect ratio more generally.

- p6 The term "body image" is used in contrast to "body representation", but it's not explained what the difference is.

- p7. "We also expected that there would be differences between the adaptation directions - such that adapting to a wider body would cause greater changes in accuracy than adapting to a narrower body". Why did you expect that? Similarly, the reason for testing the timecourse of the effect (testing at 6/12/18 minutes) isn't explained.

Results:

- I don't see any results reported about how accurate perception of body size was at baseline and how much this varied between participants. Similarly, I was a bit confused by the first paragraph of the results, where it's stated that 1-sample t-tests were conducted to test differences in perceived body size from 'accurate' (which I read to mean actual body size), but then the tests actually contrast performance following adaptation to perceived size in the baseline condition, which presumably might not be accurate.

- I wonder if the authors could comment on the size of the effects reported. For example, by expressing the apparent effect of adaptation on perceived image size in terms of change in the size of the body (e.g., in centimetres or waist size) or visual angle. Also, in Figure 4, the y-axis label is perhaps ambiguous in whether the scores are expressed as a percentage of the veridical image size or a percentage of the image size that the staircase converged on at baseline.

- Why were males/females and BSQ group included as factors in the ANOVA? What difference did you expect to see between males and females? There was a significant 4-way interaction in the ANOVA, but I could have used some more text here to highlight what is driving that interaction, or what I should be looking for in the complex plot of the data broken down across all the conditions.

Discussion:

- "We interpret these effects as altering the body representation as opposed to being a simple perceptual effect that resulted from fatiguing particular units or channels". It would be good to define a little more specifically what is meant here. The argument seems to be that the effect is not specifically visual adaptation (or lower-level visual adaptation as opposed to higher-level visual adaptation?), but relates to more abstract/high-level/multisensory representations of the body. Some of the arguments here are not convincing to me. First, visual adaptation effects can be long-lasting (e.g., one study that comes to mind reports gaze aftereffects, which are commonly interpreted in terms of changes in the sensitivity of sensory channels, lasting up to 24 hours, Kloth & Rhodes, 2016, JEP:HPP), so the 18-minute duration doesn't seem like an argument against the effects being visual or relying on changes in the sensitivity of sensory channels. Second, if you want to argue that the effects being attractive rather than repulsive is significant, it would be good to say something about what the underlying mechanism of an attractive aftereffect might be. If it's not 'fatigue' of sensory channels causing the effect of adaptation, then what else might it be? Third, it seems like the kind of approach of the cross-adaptation studies that you refer to later on (adapting to bodies vs rectangles) would be a way to test this question of how high vs low-level the effects are. e.g., if adapting to a distorted image of one's own body also

affected the perceived width of non-body objects, then this might suggest it is a lower-level visual effect.

- The manipulation is the horizontal stretch of the image (with the aspect ratio changing). So in the larger adapter image, the face features are stretched unnaturally in the horizontal dimension, as well as the rest of the body. There is a literature on adaptation to faces distorted in this kind of way (e.g., some of Mike Webster's work; 'figural aftereffects in face perception'). There also appears to be research on adaptation to the aspect ratio of objects more generally (Storrs & Arnold, 2017; <https://doi.org/10.1037/xhp0000292>). I wonder if you want to connect your work to this, as it seems closely related to the image manipulation you are using. Could your results be explained by adaptation to the aspect ratio of a familiar object? Or adaptation to the configuration of face features?

Reviewer: 2

Comments to the Author(s)

This manuscript describes an experiment showing the effect on body representations caused by exposure to full-height body stimuli depicting the participants own body digitally widened or narrowed by 20%. Participants perform a QUEST-directed 2AFC, choosing which of 2 test stimuli appear most accurate, and curve fitting establishes the width of the stimulus that matches participants' perception of their own body. While this experiment is interesting and topical, connecting with a recent surge of articles on this subject, there are several problems that need to be addressed.

MAJOR ISSUES

The first concerns definitions of the essential concepts for this study. While the title refers to "the internal representation of the body," and the abstract mentions "the implicit representation of the body", the meaning of these phrases is never defined. On pg 6, line 18, the authors make it clear that it's not the same as the body image (which is also not defined, even though this phrase has been used to mean various different things in the literature - please fix this), but don't explain in what way the two are different. It doesn't seem to refer to the current perceptual "readout" either (e.g. the "online" representation of the stimulus being viewed, such as the adaptation body or the test stimulus). (Please also explain what is meant by "central effects" - pg 6 line 14.). Without these definitions, it is difficult to fully comprehend what the thrust of the paper is, but it seems that it may be similar to recent work by Ambroziak et al., who distinguished the possible effect of adaptation on the perception of the test image vs. the effects on "internal body image". Clearly the authors have read this study, yet it is only given a token citation, saying "Ambroziak et al. (2019) used avatar images presented on a small computer screen". Their approach and findings are not covered. This is surprising, given that the two papers seem to address at least similar issues.

Another recent study to discuss the effects of adaptation on different body representations is Brooks et al. (2021). This article also relates to the description of the pattern of results, which is at odds with most adaptation papers. In the current study, the authors write: "Generally, after participants view a thinner picture of themselves, it leads them to judge their perceived body size to also be thinner...". In fact, the opposite is true (see Hummel et al, 2012; Brooks et al., 2016 already cited, plus Bould et al., 2018 & 2020 not yet cited). The description of the results as overestimation or underestimation is also discussed in Brooks et al (2021) and relates directly to the issue of what kind of representation is being affected by the process of adaptation. This problem affects many parts of the manuscript, including the description of results and discussion, e.g. on pg 17 lines 16-22. The results of previous body adaptation experiments show repulsion as described here (and as in low-level adaptation experiments, plus face adaptation results), and I believe that the current results may be in line with these (although given the problems with the

methods, I can't be sure). If the results of the current study are out of step with previous body adaptation studies, then an explanation for this discrepancy needs to be given.

The second major issue concerns the methodology, the details and rationale of which needs to be explained in full.

There is insufficient detail on the procedure and data analysis. For example, we are told "Participants identified which interval contained the image that most closely matched their perception of their own body", yet the Data Analysis section states: "To visualize and confirm the QUEST's performance, the participant's decision (0 for smaller, 1 for bigger) was plotted against the distortion used for each trial...". It is not clear how responses of "distorted image"/"undistorted image" are transformed into "smaller"/"larger". Further, it is not clear why a logistic function is used for curve-fitting when QUEST's underlying assumption is that performance reflects an underlying Weibull function. In any case, an example psychometric function fit might help readers to better understand this stage of the experiment.

Further, the rationale of the method is unclear. The authors state: "In order to assess this, we used the novel psychophysical method described in our previous studies (D'Amour & Harris, 2017, 2019, 2020). This method, since it does not involve participants ever actually seeing the image-size that they judge as their own, gives us a direct estimate of their body representation as opposed to their body image which is vulnerable to many cognitive factors." Given that the QUEST procedure involves the presentation of many different sized bodies, it seems highly likely that each participant will see the image size that they judge as their own on several occasions. Even if they did not see a body stimulus with this width, why would this mean that they're judging the "body representation" (again - this needs to be very clearly defined)? The authors need to spell this out.

Ambroziak et al. (2019) are clear that adaptation affects perception of the test images, not the stored body image. If this is true, then in the current study, adaptation should affect the perception of both of the test images approximately equally, and hence no effect should be shown. For example, after adapting to widened bodies, both test images should look thinner than they really are. This would predict that there should be no measured effects of adaptation, unless the adaptation affects the appearance of one image more than the other. It makes me wonder what the reported effects really represent.

It is also interesting that this method is described as "novel" when it appears very similar to the method used by Sekunova et al. (2013) but I can't be certain due to the lack of details. This paper should be cited.

I would also recommend greater consistency of phrasing throughout the manuscript.

Occasionally, adaptation is referred to instead as "exposure" and sometimes as "feedback". The latter example is particularly confusing, as it does not seem to be feedback in the conventional sense, unless the participant is being (deceptively) told "this is your actual body size", which I don't believe they are (please confirm).

Adaptors should either be described as bigger/smaller (as in figures) or wider/narrower, but not both.

MINOR ISSUES

Pg 3, line 21 onwards. See also Zopf et al. 2021, who showed that visual adaptation caused an aftereffect of body size but did not affect perceived tactile distance.

Pg 6 line 1. "factor" should be "factors"

Pg 6 Line 10. There was no adaptation in the Urdapilleta et al., 2010 study.

Pg 5 line 12 "How much of this is a simple visual after effect, comparable to low-level adaptation..." This statement is strange, given that pg 18 cover the evidence that body adaptation is high level. Also, see Pg 17, line 23

Pg 6, line 7 “We also expected that there would be differences between the adaptation directions – such that adapting to a wider body would cause greater changes in accuracy than adapting to a narrower body.” Why? This hypothesis needs to be justified.

Pg9 line 9 2.2.4. Altering visual feedback with images – This section seems to be procedure, not Materials/Stimuli

Pg 9 Figure 2. Here, the height of the images seems to vary between the 3 images. Please explain.

Pg 11, line 5. “separated by a blank white screen.” For how long?

Pg 15 Figure 4. Remove title from graph. Legend should match text. Don't plot a point at zero,

Pg 16 line 8. These findings clearly answer the question of whether visually manipulating body representation can alter accuracy values...” I would disagree with this. The idea that representation has been manipulated is inferred, given the effects measured. It is not a given, whose effects are then assessed.

Pg 16 line 13” We interpret these effects as altering the body representation as opposed to being a simple perceptual effect that resulted from fatiguing particular units or channels.” This comment seems outdated. It has been a very long time since the theory that adaptation is due to fatigue was popular or considered credible. More recent accounts suggest recalibration. A reference more recent than Blakemore 1970 should probably be given.

Pg 16, line 23 “If our adaptation effects were due only to low-level visual perception adaptation mechanisms then everyone should be similar and show comparable levels of adaptation.” Again, it is clear that body adaptation is high level. But even so, the authors need to justify this statement. Why should “everyone be similar”?

Pg 18 line 7. “The body representation’s susceptibility to change after such brief experience suggests body representation may be even more plastic than previously thought.” Given that there have been approximately 20 studies to date that demonstrate body size adaptation, should we really be surprised?

Bould, H., Carnegie, R., Allward, H., Bacon, E., Lambe, E., Sapstead, M., ... & Munafò, M. R. (2018). Effects of exposure to bodies of different sizes on perception of and satisfaction with own body size: two randomized studies. *Royal Society open science*, 5(5), 171387.

Bould, H., Noonan, K., Penton-Voak, I., Skinner, A., Munafò, M. R., Park, R. J., ... & Harmer, C. J. (2020). Does repeatedly viewing overweight versus underweight images change perception of and satisfaction with own body size? *Royal Society Open Science*, 7(4), 190704.

Brooks, K. R., Stevenson, R. J., & Stephen, I. D. (2021). Over or Under? Mental Representations and the Paradox of Body Size Estimation. *Frontiers in Psychology*, 12.

Sekunova, A., Black, M., Parkinson, L., & Barton, J. J. (2013). Viewpoint and pose in body-form adaptation. *Perception*, 42(2), 176-186.

Zopf, R., Kosourkhina, V., Brooks, K. R., Polito, V. & Stephen, I. D. (2021). Visual body size adaptation and estimation of tactile distance. *British Journal of Psychology*. doi: 10.1111/bjop.12514

===PREPARING YOUR MANUSCRIPT===

===PREPARING YOUR REVISION IN SCHOLARONE===

<https://royalsociety.org/journals/authors/author-guidelines/#supplementary-material> to include a suitable title and informative caption. An example of appropriate titling and captioning may be found at https://figshare.com/articles/Table_S2_from_Is_there_a_trade-off_between_peak_performance_and_performance_breadth_across_temperatures_for_aerobic_scooping_in_teleost_fishes_/3843624.

Author's Response to Decision Letter for (RSOS-210722.R0)

See Appendix A.

RSOS-210722.R1 (Revision)

Review form: Reviewer 1

Is the manuscript scientifically sound in its present form?

Yes

Are the interpretations and conclusions justified by the results?

No

Is the language acceptable?

Yes

Do you have any ethical concerns with this paper?

No

Have you any concerns about statistical analyses in this paper?

No

Recommendation?

Accept with minor revision (please list in comments)

Comments to the Author(s)

Thank you to the authors for revising the manuscript. The added details in the Method were helpful for understanding the direction of the effects.

This new sentence in the Discussion sums up the explanation of the results that seems most convincing to me: "...after exposure to a wider body, the test photographs could have appeared narrower, meaning a wider one would need to have been chosen to match their own body representation", i.e., a repulsive visual aftereffect. This contrasts with the author's preferred explanation, which seems to be that adaptation is not affecting perception of the images shown in each trial but is rather affecting an internal sense of the body that is independent of these.

This is relevant to the previous paragraph in the Discussion, where the direction of the effect is cited as a reason to think that it is not a visual aftereffect. Because, actually, the direction of the effect *is* consistent with a typical repulsive visual aftereffect. Also, the sentence "Our technique of not asking our participants to view their own image makes this less likely" doesn't seem relevant, because one would expect a visual aftereffect to act on the perceived size of an image regardless of whether that size matches the participant's own body size or not.

- A minor thing I found slightly confusing is that it's stated a couple of times that with the images shown in each trial "neither ... matches their actual idea of themselves", but actually the reference image shown in each trial is an undistorted image that matches pretty closely with the viewer's sense of their own body size at baseline (though less so after adaptation).

Review form: Reviewer 2

Is the manuscript scientifically sound in its present form?

No

Are the interpretations and conclusions justified by the results?

No

Is the language acceptable?

Yes

Do you have any ethical concerns with this paper?

No

Have you any concerns about statistical analyses in this paper?

No

Recommendation?

Major revision is needed (please make suggestions in comments)

Comments to the Author(s)

Thank you for the opportunity to take a look at the authors' responses to my first review. Although I acknowledge that they have amended various parts of the manuscript, I regret to inform that I am no closer to recommending publication than I was after the first review. The major concerns at the time of the first review concerned the definition of "representation"; the details of the methodology, and the assertion that this method "gives us a direct estimate of their body representation". The first and second problems interfered with my ability to fully critique the latter. The methods used for this study are now better explained, and the essential

concepts for this study (the “internal representation of the body”) are better defined (“we consider it a kind of memory”). However, now that I understand the concepts and methods, I have more serious concerns with the assertion that the method measures changes in the representation of your own body in memory.

In the revised document, it is again stated: “Thus, this method gives us a best estimate of their body representation as ultimately in the procedure they are forced to rely on accessing their internal body representation to decide which of two photographs neither of which actually matches this representation is closest to it.”

In the reply to reviewer #1, it is stated: “Our psychophysical task is unique because the subject is never shown the actual image that matches their own. Instead, they need to compare their internal idea of their own body size with the presented image. We then end up with two images, neither of which actually matches their representation but from which the size of their representation can be induced. We are not able to say at which level these changes occur – only that they cause the internal representation to be altered. That is the main contribution of this paper – which we emphasize in the title.”

Firstly, the authors now acknowledge that the participants in fact do see an actual image that matches their body, but “only transiently”. But my re-reading of the (now clearer) methods section makes it clear that they in fact do see an image that matches their body size on every single trial. From section 2.3: “Each trial consisted of two 1.5s intervals – one interval containing an undistorted natural-size image of themselves (the reference image) and one interval containing a distorted image.”

Apart from that, even if the participants hadn’t seen such an image, it is not clear why this should somehow inoculate them from experiencing any kind of conventional high-level perceptual adaptation that affects the appearance of the visual stimuli shown on screen.

It is clear that any judgement in this manuscript must involve the comparison of the visual stimuli (in this case both the reference and test image) with the mental representation of the participant’s own body (see Brooks et al., 2021). It is of course possible that biased results could arise from changes in the perceived size of the visual images on screen, or a change in the size of the mental representation. Just as in low-level aftereffects, or in high level aftereffects such as those that have been shown to affect faces, visual adaptation is assumed to affect the former. Why the authors assert that their adaptation instead affects the latter is unclear.

In the revised discussion, the authors now acknowledge that they “appreciate that this could be a consequence of the photographs themselves becoming perceptually distorted”. Why this is an afterthought, and not the primary assumption is not clear. Regardless, given the acknowledged possibility that the results might not involve any distortion of the body representation in memory, the authors must not make statements to the effect that the body representation has changed as if it were a fact throughout the document (e.g., title, abstract, conclusions, etc.)

A new section in the discussion reads: “A recent review by Brooks et al. (2020) reviewed several key points surrounding adaptation in body size and provided support for high-level adaption of body representation.” That is not my reading of Brooks et al., 2020. Their discussion of these studies is very much in the tradition of visual adaptation (low and high level) with the core assumption that it affects the current perceptual readout, not the stored representation of the body (see also Brooks et al, 2021). Also, in this section, the authors conflate the issue of low-level vs high level adaptation with the issue of whether exposure to body stimuli affect the perceptual readout or the stored representation of their own body. These two are entirely separate issues, yet this confusion is found in several places in the manuscript. As far as I can see, there is no doubt that low-level (i.e. retinotopic) adaptation is not responsible for body size aftereffects, but to say that this means that it must affect the stored representation does not flow logically.

Further, the fact that the results were somewhat variable as a function of gender and body dissatisfaction may argue against low-level retinotopic visual aftereffects, but it does not argue against high level visual aftereffects. Note that attention is well known to moderate body aftereffect size. Further, Stephen et al., has previously shown the effect of attention on body size and shape aftereffects, as moderated by body dissatisfaction.

However, all is not lost. Although the current report of an attractive aftereffect caused by a change on the stored representation is controversial and out of step with previous research, when viewed a distortion of the current visual percept (i.e., of the reference and test stimulus) the results make sense perfect. In this perspective, the aftereffects shown here are negative (i.e., the visual stimuli look distorted in the opposite direction to the adaptors), as they are in all other body aftereffect studies that I am aware of. (Note the new text: “after exposure to a wider body, the test photographs could have appeared narrower, meaning a wider one would need to have been chosen to match their own body representation.”) In this case, the current study is a neat demonstration that the conventional aftereffects of body size can be produced from life-size stimuli (never been shown before) and can last up to 18minutes (also a novel finding), with some variation associated with gender and body dissatisfaction (this is not so novel, but is interesting nevertheless). Why not drop the controversial claims about “body representation” and simply write up the results to emphasize these useful observations? Although this would involve substantial changes to the manuscript, I believe that such a manuscript would make a valuable contribution to the literature.

Decision letter (RSOS-210722.R1)

Dear Dr D'Amour

The Editors assigned to your paper RSOS-210722.R1 "Changes in the internal representation of the body following exposure to distorted self-body images" have now received comments from reviewers and would like you to revise the paper in accordance with the reviewer comments and any comments from the Editors. Please note this decision does not guarantee eventual acceptance.

Please submit your revised manuscript and required files (see below) no later than 21 days from today's (ie 15-Nov-2021) date. Note: the ScholarOne system will 'lock' if submission of the revision is attempted 21 or more days after the deadline. If you do not think you will be able to meet this deadline please contact the editorial office immediately.

Please note article processing charges apply to papers accepted for publication in Royal Society Open Science (<https://royalsocietypublishing.org/rsos/charges>). Charges will also apply to papers transferred to the journal from other Royal Society Publishing journals, as well as papers submitted as part of our collaboration with the Royal Society of Chemistry

(<https://royalsocietypublishing.org/rsos/chemistry>). Fee waivers are available but must be requested when you submit your revision (<https://royalsocietypublishing.org/rsos/waivers>).

on behalf of Dr Isabelle Mareschal (Associate Editor) and Essi Viding (Subject Editor)
openscience@royalsociety.org

Associate Editor Comments to Author (Dr Isabelle Mareschal):

Both reviewers are not convinced that the effect is one of adaptation of the internal representation rather than perceptual adaptation. There are also still confusions about the method itself, notably about whether participants view an image of their internal representation with the distorted image or not (I agree with both reviewers that this seems to be the case although the authors say it isn't). Unfortunately it is not possible to work out exactly what was done from the method section. I don't want to have a protracted back and forth with the reviewers so please ensure you answer all questions clearly in your reply and make changes in the manuscript. Notably the description of the procedure (e.g. is the "reference" different from the representation?) and the corresponding evidence for adaptation of body representation per se.

Reviewer comments to Author:

Reviewer: 1

Comments to the Author(s)

Thank you to the authors for revising the manuscript. The added details in the Method were helpful for understanding the direction of the effects.

This new sentence in the Discussion sums up the explanation of the results that seems most convincing to me: "...after exposure to a wider body, the test photographs could have appeared narrower, meaning a wider one would need to have been chosen to match their own body representation", i.e., a repulsive visual aftereffect. This contrasts with the author's preferred explanation, which seems to be that adaptation is not affecting perception of the images shown in each trial but is rather affecting an internal sense of the body that is independent of these.

This is relevant to the previous paragraph in the Discussion, where the direction of the effect is cited as a reason to think that it is not a visual aftereffect. Because, actually, the direction of the effect *is* consistent with a typical repulsive visual aftereffect. Also, the sentence "Our technique of not asking our participants to view their own image makes this less likely" doesn't seem relevant, because one would expect a visual aftereffect to act on the perceived size of an image regardless of whether that size matches the participant's own body size or not.

- A minor thing I found slightly confusing is that it's stated a couple of times that with the images shown in each trial "neither ... matches their actual idea of themselves", but actually the reference image shown in each trial is an undistorted image that matches pretty closely with the viewer's sense of their own body size at baseline (though less so after adaptation).

Reviewer: 2

Comments to the Author(s)

Thank you for the opportunity to take a look at the authors' responses to my first review.

Although I acknowledge that they have amended various parts of the manuscript, I regret to inform that I am no closer to recommending publication than I was after the first review.

The major concerns at the time of the first review concerned the definition of "representation"; the details of the methodology, and the assertion that this method "gives us a direct estimate of their body representation". The first and second problems interfered with my ability to fully critique the latter. The methods used for this study are now better explained, and the essential concepts for this study (the "internal representation of the body") are better defined ("we consider it a kind of memory"). However, now that I understand the concepts and methods, I have more serious concerns with the assertion that the method measures changes in the representation of your own body in memory.

In the revised document, it is again stated: "Thus, this method gives us a best estimate of their body representation as ultimately in the procedure they are forced to rely on accessing their internal body representation to decide which of two photographs neither of which actually matches this representation is closest to it."

In the reply to reviewer #1, it is stated: "Our psychophysical task is unique because the subject is never shown the actual image that matches their own. Instead, they need to compare their internal idea of their own body size with the presented image. We then end up with two images, neither of which actually matches their representation but from which the size of their representation can be induced. We are not able to say at which level these changes occur - only that they cause the internal representation to be altered. That is the main contribution of this paper - which we emphasize in the title."

Firstly, the authors now acknowledge that the participants in fact do see an actual image that matches their body, but "only transiently". But my re-reading of the (now clearer) methods section makes it clear that they in fact do see an image that matches their body size on every single trial. From section 2.3: "Each trial consisted of two 1.5s intervals - one interval containing an undistorted natural-size image of themselves (the reference image) and one interval containing a distorted image."

Apart from that, even if the participants hadn't seen such an image, it is not clear why this should somehow inoculate them from experiencing any kind of conventional high-level perceptual adaptation that affects the appearance of the visual stimuli shown on screen.

It is clear that any judgement in this manuscript must involve the comparison of the visual stimuli (in this case both the reference and test image) with the mental representation of the participant's own body (see Brooks et al., 2021). It is of course possible that biased results could arise from changes in the perceived size of the visual images on screen, or a change in the size of the mental representation. Just as in low-level aftereffects, or in high level aftereffects such as those that have been shown to affect faces, visual adaptation is assumed to affect the former. Why the authors assert that their adaptation instead affects the latter is unclear.

In the revised discussion, the authors now acknowledge that they "appreciate that this could be a consequence of the photographs themselves becoming perceptually distorted". Why this is an afterthought, and not the primary assumption is not clear. Regardless, given the acknowledged possibility that the results might not involve any distortion of the body representation in memory, the authors must not make statements to the effect that the body representation has changed as if it were a fact throughout the document (e.g., title, abstract, conclusions, etc.)

A new section in the discussion reads: "A recent review by Brooks et al. (2020) reviewed several key points surrounding adaptation in body size and provided support for high-level adaption of body representation." That is not my reading of Brooks et al., 2020. Their discussion of these studies is very much in the tradition of visual adaptation (low and high level) with the core assumption that it affects the current perceptual readout, not the stored representation of the body (see also Brooks et al, 2021). Also, in this section, the authors conflate the issue of low-level vs high level adaptation with the issue of whether exposure to body stimuli affect the perceptual

readout or the stored representation of their own body. These two are entirely separate issues, yet this confusion is found in several places in the manuscript. As far as I can see, there is no doubt that low-level (i.e. retinotopic) adaptation is not responsible for body size aftereffects, but to say that this means that it must affect the stored representation does not flow logically.

Further, the fact that the results were somewhat variable as a function of gender and body dissatisfaction may argue against low-level retinotopic visual aftereffects, but it does not argue against high level visual aftereffects. Note that attention is well known to moderate body aftereffect size. Further, Stephen et al., has previously shown the effect of attention on body size and shape aftereffects, as moderated by body dissatisfaction.

However, all is not lost. Although the current report of an attractive aftereffect caused by a change on the stored representation is controversial and out of step with previous research, when viewed a distortion of the current visual percept (i.e., of the reference and test stimulus) the results make sense perfect. In this perspective, the aftereffects shown here are negative (i.e., the visual stimuli look distorted in the opposite direction to the adaptors), as they are in all other body aftereffect studies that I am aware of. (Note the new text: "after exposure to a wider body, the test photographs could have appeared narrower, meaning a wider one would need to have been chosen to match their own body representation.") In this case, the current study is a neat demonstration that the conventional aftereffects of body size can be produced from life-size stimuli (never been shown before) and can last up to 18minutes (also a novel finding), with some variation associated with gender and body dissatisfaction (this is not so novel, but is interesting nevertheless). Why not drop the controversial claims about "body representation" and simply write up the results to emphasize these useful observations? Although this would involve substantial changes to the manuscript, I believe that such a manuscript would make a valuable contribution to the literature.

===PREPARING YOUR MANUSCRIPT===

If you have been asked to revise the written English in your submission as a condition of publication, you must do so, and you are expected to provide evidence that you have received language editing support. The journal would prefer that you use a professional language editing service and provide a certificate of editing, but a signed letter from a colleague who is a fluent speaker of English is acceptable. Note the journal has arranged a number of discounts for authors

using professional language editing services
(<https://royalsociety.org/journals/authors/benefits/language-editing/>).

===PREPARING YOUR REVISION IN SCHOLARONE===

<https://royalsociety.org/journals/authors/author-guidelines/#supplementary-material> to include a suitable title and informative caption. An example of appropriate titling and captioning may be found at https://figshare.com/articles/Table_S2_from_Is_there_a_trade-

off_between_peak_performance_and_performance_breadth_across_temperatures_for_aerobic_sc
ope_in_teleost_fishes_/3843624.

Author's Response to Decision Letter for (RSOS-210722.R1)

See Appendix B.

RSOS-210722.R2

Review form: Reviewer 1

Is the manuscript scientifically sound in its present form?

Yes

Are the interpretations and conclusions justified by the results?

No

Is the language acceptable?

Yes

Do you have any ethical concerns with this paper?

No

Have you any concerns about statistical analyses in this paper?

No

Recommendation?

Accept with minor revision (please list in comments)

Comments to the Author(s)

Thank you to the authors for considering my comments.

It still seems to me that, following adaptation, if the participant's internal representation of their own body size is unaffected, but their perception of the body width in the images presented on-screen is influenced by adaptation, they would show the same results to those reported here. (e.g., following adaptation to wide images, participants would select a wider test photo as being more similar to their internal representation, because they would see that test photo as being narrower than it really is).

The point about 'not asking participants to view their own image' doesn't seem relevant to this. The arguments against it being a low-level visual aftereffect don't seem to preclude it from being a high-level visual aftereffect. The results appear equally consistent with a body-shape-specific,

repulsive effect of adaptation on the appearance of seen bodies, rather than the attractive effect on the internal representation of the body that the authors describe.

I didn't follow why the slopes of the psychometric function would be diagnostic about whether the results reflect a visual aftereffect or not. Perhaps the logic could be spelt out more here.

Review form: Reviewer 2

Is the manuscript scientifically sound in its present form?

No

Are the interpretations and conclusions justified by the results?

No

Is the language acceptable?

Yes

Do you have any ethical concerns with this paper?

No

Have you any concerns about statistical analyses in this paper?

Yes

Recommendation?

Reject

Comments to the Author(s)

I have now had time to peruse the second set of revisions made by the authors, and the responses to reviewers.

Overall, the authors have chosen to double down on their original assertions, principally that "this method gives us a best estimate of their body representation as ultimately in the procedure they are forced to rely on accessing their internal body representation" and that "visual feedback impacts the internal representation of body size". I remain unconvinced.

In my previous review, I noted: "it is not clear why this [the still-disputed claim that the participants never see an image that matches their memory representation of their own body] should somehow inoculate them from experiencing any kind of conventional high-level perceptual adaptation that affects the appearance of the visual stimuli shown on screen." For this, the authors had no answer (unless it is the words "We agree completely, and hope that our edits make this clear" although to be fair, I doubt that this was intended to be a direct response to this particular concern).

Repeatedly, in responses to reviewers, and in the revised manuscript, the authors try to convince us that the effects they show are unlikely to be due to low-level (i.e., retinotopic) adaptation (note that in the revision the phrase "low-level" has been added in 3 locations). Further, new data on psychometric function slopes are provided. The authors claim: "If our results could be explained as "repulsive visual after effect" then we would expect everything in the image to be scaled up and the slopes of the psychometric functions to be correspondingly affected also (adapt wider, slopes shallower; adapt narrow, slopes steeper)." Unfortunately, the lack of statistically significant differences between slopes for each condition is unconvincing for several reasons.

First, the QUEST procedure used here may be effective for determining a psychometric function mid-point, but it is far from optimal for determining slope. Some function parameters are calculated from as few as 16 trials (see figure 4). Many more trials, targeted at upper and lower points of the curve, would be required to achieve any kind of reliable slope measurement. On top of this, null results (especially those produced by noisy measurements) are never convincing.

That low-level adaptation is not responsible for the effects here is not disputed by anyone. That body adaptation is a high-level effect has been shown by several studies (including Hummel et al., of which the authors are aware). Yet as I stated previously, this does not leave effects on memory as the only logical alternative. High-level perceptual (not memory) aftereffects have been widely reported for over 20 years, initially in the face perception literature, and are the most obvious explanation here. In my previous review I suggested that the authors report their interesting effects in this more conventional framework, but sadly they have declined this opportunity. To assert that the effects shown here are due to effects on stored representations in memory would require additional experiments. Without clear evidence, such claims are speculative at best. As in my second review, I maintain that the assumption that memory must be affected must not form the framework for the entire paper, and certainly cannot appear in the title, abstract or conclusions.

I appreciate the hard work that has gone into this paper, but given that the authors' have declined to take my previous two sets of comments on board, I regret that I am unable to recommend publication of this manuscript.

Decision letter (RSOS-210722.R2)

Dear Dr D'Amour,

On behalf of the Editors, we are pleased to inform you that your Manuscript RSOS-210722.R2 "Changes in the internal representation of the body following exposure to distorted self-body images" has been accepted for publication in Royal Society Open Science subject to minor revision in accordance with the referees' reports. Please find the referees' comments along with any feedback from the Editors below my signature.

Please submit your revised manuscript and required files (see below) no later than 7 days from today's (ie 28-Feb-2022) date. Note: the ScholarOne system will 'lock' if submission of the revision is attempted 7 or more days after the deadline. If you do not think you will be able to meet this deadline please contact the editorial office immediately.

Please note article processing charges apply to papers accepted for publication in Royal Society Open Science (<https://royalsocietypublishing.org/rsos/charges>). Charges will also apply to papers transferred to the journal from other Royal Society Publishing journals, as well as papers submitted as part of our collaboration with the Royal Society of Chemistry

(<https://royalsocietypublishing.org/rsos/chemistry>). Fee waivers are available but must be requested when you submit your revision (<https://royalsocietypublishing.org/rsos/waivers>).

on behalf of Dr Isabelle Mareschal (Associate Editor) and Essi Viding (Subject Editor)
openscience@royalsociety.org

Associate Editor Comments to Author (Dr Isabelle Mareschal):

Reviewer 2 remains unconvinced that your result is solely attributable to adaptation of the internal representation. Although altering the internal representation may play a role, it is not clear to me that this is the only explanation and/or that other factors may not also be involved. However, the paper does present an interesting method and results, and it has been through a few rounds of review now, so in the interest of time, I will make a final decision on the paper. I will not send it back to review but I need you to provide a more nuanced framing of the interpretation / discussion that leaves room for a role / influence of other (higher level) factors. If you can do this please resubmit your paper with a clear description what you changed and where for me to evaluate.

Reviewer comments to Author:

Reviewer: 1

Comments to the Author(s)

Thank you to the authors for considering my comments.

It still seems to me that, following adaptation, if the participant's internal representation of their own body size is unaffected, but their perception of the body width in the images presented on-screen is influenced by adaptation, they would show the same results to those reported here. (e.g., following adaptation to wide images, participants would select a wider test photo as being more similar to their internal representation, because they would see that test photo as being narrower than it really is).

The point about 'not asking participants to view their own image' doesn't seem relevant to this. The arguments against it being a low-level visual aftereffect don't seem to preclude it from being a high-level visual aftereffect. The results appear equally consistent with a body-shape-specific, repulsive effect of adaptation on the appearance of seen bodies, rather than the attractive effect on the internal representation of the body that the authors describe.

I didn't follow why the slopes of the psychometric function would be diagnostic about whether the results reflect a visual aftereffect or not. Perhaps the logic could be spelt out more here.

Reviewer: 2

Comments to the Author(s)

I have now had time to peruse the second set of revisions made by the authors, and the responses to reviewers.

Overall, the authors have chosen to double down on their original assertions, principally that “this method gives us a best estimate of their body representation as ultimately in the procedure they are forced to rely on accessing their internal body representation” and that “visual feedback impacts the internal representation of body size”. I remain unconvinced.

In my previous review, I noted: “it is not clear why this [the still-disputed claim that the participants never see an image that matches their memory representation of their own body] should somehow inoculate them from experiencing any kind of conventional high-level perceptual adaptation that affects the appearance of the visual stimuli shown on screen.” For this, the authors had no answer (unless it is the words “We agree completely, and hope that our edits make this clear” although to be fair, I doubt that this was intended to be a direct response to this particular concern).

Repeatedly, in responses to reviewers, and in the revised manuscript, the authors try to convince us that the effects they show are unlikely to be due to low-level (i.e., retinotopic) adaptation (note that in the revision the phrase “low-level” has been added in 3 locations). Further, new data on psychometric function slopes are provided. The authors claim: “If our results could be explained as “repulsive visual after effect” then we would expect everything in the image to be scaled up and the slopes of the psychometric functions to be correspondingly affected also (adapt wider, slopes shallower; adapt narrow, slopes steeper).” Unfortunately, the lack of statistically significant differences between slopes for each condition is unconvincing for several reasons. First, the QUEST procedure used here may be effective for determining a psychometric function mid-point, but it is far from optimal for determining slope. Some function parameters are calculated from as few as 16 trials (see figure 4). Many more trials, targeted at upper and lower points of the curve, would be required to achieve any kind of reliable slope measurement. On top of this, null results (especially those produced by noisy measurements) are never convincing.

That low-level adaptation is not responsible for the effects here is not disputed by anyone. That body adaptation is a high-level effect has been shown by several studies (including Hummel et al., of which the authors are aware). Yet as I stated previously, this does not leave effects on memory as the only logical alternative. High-level perceptual (not memory) aftereffects have been widely reported for over 20 years, initially in the face perception literature, and are the most obvious explanation here. In my previous review I suggested that the authors report their interesting effects in this more conventional framework, but sadly they have declined this opportunity. To assert that the effects shown here are due to effects on stored representations in memory would require additional experiments. Without clear evidence, such claims are speculative at best. As in my second review, I maintain that the assumption that memory must be affected must not form the framework for the entire paper, and certainly cannot appear in the title, abstract or conclusions.

I appreciate the hard work that has gone into this paper, but given that the authors’ have declined to take my previous two sets of comments on board, I regret that I am unable to recommend publication of this manuscript.

===PREPARING YOUR MANUSCRIPT===

one version should clearly identify all the changes that have been made (for instance, in coloured highlight, in bold text, or tracked changes);

===PREPARING YOUR REVISION IN SCHOLARONE===

- Any electronic supplementary material (ESM).
- If you are requesting a discretionary waiver for the article processing charge, the waiver form must be included at this step.
- If you are providing image files for potential cover images, please upload these at this step, and inform the editorial office you have done so. You must hold the copyright to any image provided.
- A copy of your point-by-point response to referees and Editors. This will expedite the preparation of your proof.

- Ensure that your data access statement meets the requirements at <https://royalsociety.org/journals/authors/author-guidelines/#data>. You should ensure that you cite the dataset in your reference list. If you have deposited data etc in the Dryad repository, please only include the 'For publication' link at this stage. You should remove the 'For review' link.
- If you are requesting an article processing charge waiver, you must select the relevant waiver option (if requesting a discretionary waiver, the form should have been uploaded, see 'File upload' above).
- If you have uploaded any electronic supplementary (ESM) files, please ensure you follow the guidance at <https://royalsociety.org/journals/authors/author-guidelines/#supplementary-material> to include a suitable title and informative caption. An example of appropriate titling and captioning may be found at https://figshare.com/articles/Table_S2_from_Is_there_a_trade-off_between_peak_performance_and_performance_breadth_across_temperatures_for_aerobic_scope_in_teleost_fishes_/3843624.

Author's Response to Decision Letter for (RSOS-210722.R2)

See Appendix C.

Decision letter (RSOS-210722.R3)

Dear Dr D'Amour,

It is a pleasure to accept your manuscript entitled "Changes in the perceived size of the body following exposure to distorted self-body images" in its current form for publication in Royal Society Open Science.

If you have not already done so, please ensure that you send to the editorial office an editable version of your accepted manuscript, and individual files for each figure and table included in

your manuscript. You can send these in a zip folder if more convenient. Failure to provide these files may delay the processing of your proof.

on behalf of Dr Isabelle Mareschal (Associate Editor) and Essi Viding (Subject Editor)
openscience@royalsociety.org

Appendix A

Associate Editor Comments to Author (Dr Isabelle Mareschal):

Associate Editor: 1

Comments to the Author:

Both reviewers raise issues around the theoretical rationale of the experiments (which could be made clearer by also clearly defining concepts and linking these to the relevant literature). The methodology also needs more clarification to help interpret the results. Please provide a point by point answer to all questions raised.

Reviewer comments to Author:

Reviewer: 1

Comments to the Author(s)

The general topic of the paper is really interesting: how viewing images of oneself (e.g. in the mirror) influences our perception of body size. The authors review past research indicating that visual feedback about body size can affect body representation, as well as studies showing that adaptation to images of bodies can cause (attractive) perceptual aftereffects when making judgements about one's own body size. The current study seems to differ mainly in showing participants life-sized images of their own body. Adaptation to images of oneself with a distorted horizontal width impacted subsequent judgments about the 'correct' width of these images (and these were attractive aftereffects). I didn't get much of a sense from the paper about what mechanism might underlie these effects. I found some ideas in the paper to be a bit under-explained. Specific comments below.

Introduction:

– An idea conveyed in the Introduction is that the psychophysical task that the experiment uses taps into higher-level body representation rather than lower-level visual adaptation. However, the design of the task seems like it could capture lower-level visual aftereffects perfectly well. i.e., you are adapted to a stretched or compressed image of a body, then you are tested on similar images presented in the same area of the visual field. In the Discussion, there are some arguments for why the results might reflect adaptation at a higher level, but it doesn't seem to be the case that the task itself is designed to isolate higher-level from lower-level effects, or to isolate effects specific to body representation from adaptation to object size, shape or aspect ratio more generally.

Response 1

Our psychophysical task is unique because the subject is never shown the actual image that matches their own. Instead, they need to compare their internal idea of their own body size with the presented image. We then end up with two images, neither of which actually matches their representation but from which the size of their representation can be induced. We are not able to say at which level these changes occur – only that they cause the internal representation to be altered. That is the main contribution of this paper – which we emphasize in the title.

To make our argument clearer we have made edits to the abstract, introduction and discussion.

– p6 The term “body image” is used in contrast to “body representation”, but it’s not explained what the difference is.

Response 2

We have deleted reference to “body image”.

– p7. “We also expected that there would be differences between the adaptation directions – such that adapting to a wider body would cause greater changes in accuracy than adapting to a narrower body”. Why did you expect that? Similarly, the reason for testing the timecourse of the effect (testing at 6/12/18 minutes) isn’t explained.

Response 3

This speculation was based on the clinical literature in which it is quite common for people to feel fatter than they are (in anorexia nervosa, for example) but unusual for people to feel thinner than they are. Neurotypical people are also more likely to think they are fatter than they are. References have been added to back up these points.

The time course is interesting because low-level aftereffects last usually for only a few seconds after exposure. Both these lines of thought have been explained in the last paragraph of the introduction.

Results:

– I don’t see any results reported about how accurate perception of body size was at baseline and how much this varied between participants. Similarly, I was a bit confused by the first paragraph of the results, where it’s stated that 1-sample t-tests were conducted to test differences in perceived body size from ‘accurate’ (which I read to mean actual body size), but then the tests actually contrast performance following adaptation to perceived size in the baseline condition, which presumably might not be accurate.

Response 4

Baseline accuracy for each participant was subtracted from all their other scores to provide a standard baseline reference. We have added a statement about typical baseline errors to the beginning of the results, but this is not material to our case. All statistical tests we carried out, as stated on page 12, relative to the normalized baseline. We were not concerned here as to how accurate their individual baseline scores were. There was an error in the text in which we referred to “different from accurate” as opposed to “different from baseline”. This has now been corrected.

– I wonder if the authors could comment on the size of the effects reported. For example, by expressing the apparent effect of adaptation on perceived image size in terms of change in the size of the body (e.g., in centimetres or waist size) or visual angle. Also, in Figure 4, the y-axis

label is perhaps ambiguous in whether the scores are expressed as a percentage of the veridical image size or a percentage of the image size that the staircase converged on at baseline.

Response 5

It is not possible to comment on changes of waist size or visual angle because, as confirmed in response 4, all measures were relative to each individual's baseline. An interested reader could perhaps do some calculations to approximate the measures you suggest for some typical participant. The axis is labelled "Difference from baseline" and is a percentage of that baseline value. This has been clarified further in the legend.

– Why were males/females and BSQ group included as factors in the ANOVA? What difference did you expect to see between males and females? There was a significant 4-way interaction in the ANOVA, but I could have used some more text here to highlight what is driving that interaction, or what I should be looking for in the complex plot of the data broken down across all the conditions.

Response 6

Males and females were included in the analysis because we, and we thought, the community, might expect sex differences. Our rationale for this expectation has now been added to the introduction. We have now added a detailed account of post-hoc follow up statistics to investigate what was driving the interaction.

Discussion:

– “We interpret these effects as altering the body representation as opposed to being a simple perceptual effect that resulted from fatiguing particular units or channels”. It would be good to define a little more specifically what is meant here. The argument seems to be that the effect is not specifically visual adaptation (or lower-level visual adaptation as opposed to higher-level visual adaptation?), but relates to more abstract/high-level/multisensory representations of the body. Some of the arguments here are not convincing to me. First, visual adaptation effects can be long-lasting (e.g., one study that comes to mind reports gaze aftereffects, which are commonly interpreted in terms of changes in the sensitivity of sensory channels, lasting up to 24 hours, Kloth & Rhodes, 2016, JEP:HPP), so the 18-minute duration doesn't seem like an argument against the effects being visual or relying on changes in the sensitivity of sensory channels.

Response 7

It is true that some after-effects can last for a long time such as the Kloth and Rhodes example you mention, but these may be rather special cases. We felt that simple spatial frequency aftereffects (after gazing at low or high spatial frequency gratings, for example) or the waterfall after effect (now added) were more directly comparable to the present study. Such effects are known to be typically short lived and in the negative direction (opposite to the adapting stimulus). See also response 28 below.

Second, if you want to argue that the effects being attractive rather than repulsive is significant, it would be good to say something about what the underlying mechanism of an attractive aftereffect might be. If it's not 'fatigue' of sensory channels causing the effect of adaptation, then what else might it be?

Response 8

Thank you for providing an opportunity to speculate. Our model for what is underlying our effects is that the person, seeing themselves to have a certain dimension, adjust their internal representation to match. We have now made this concept more explicit in the discussion.

Third, it seems like the kind of approach of the cross-adaptation studies that you refer to later on (adapting to bodies vs rectangles) would be a way to test this question of how high vs low-level the effects are. e.g., if adapting to a distorted image of one's own body also affected the perceived width of non-body objects, then this might suggest it is a lower-level visual effect.

Response 9

A concern here is that we also have internal representations of non-body objects (e.g., a coke can) and can draw them reasonably accurately from memory. Repeatedly seeing wider coke cans would presumably alter our idea of the normal size of a coke can in the same way. It is only by contrasting our results with the results of formless patterns (such as gratings) that a control can be carried out. And such experiments have been reported frequently in the literature to which we gave one representative example (and have added another). Hummel et al. 2012 give other perhaps more directly relevant examples.

– The manipulation is the horizontal stretch of the image (with the aspect ratio changing). So in the larger adapter image, the face features are stretched unnaturally in the horizontal dimension, as well as the rest of the body. There is a literature on adaptation to faces distorted in this kind of way (e.g., some of Mike Webster's work; 'figural aftereffects in face perception'). There also appears to be research on adaptation to the aspect ratio of objects more generally (Storrs & Arnold, 2017; <https://doi.org/10.1037/xhp0000292>). I wonder if you want to connect your work to this, as it seems closely related to the image manipulation you are using. Could your results be explained by adaptation to the aspect ratio of a familiar object? Or adaptation to the configuration of face features?

Response 10

Adaptation to object aspect ratio is another example of a relatively low-level effect in which the after-effects are generally negative (exposing to wider objects leads to narrower perceptions). We do not feel this literature is particularly pertinent here although in general it supports our case about the difference between low- and high-level effects. The face experiments of Webster are more closely linked to the studies by Stephen et al. (2016) <https://doi.org/10.3389/fpsyg.2016.01223> which attempted to modulate the effect of body size distortions by drawing people's attention to certain features. This turned out to have no significant effect on the representation of the body.

Reviewer: 2

Comments to the Author(s)

This manuscript describes an experiment showing the effect on body representations caused by exposure to full-height body stimuli depicting the participants own body digitally widened or narrowed by 20%. Participants perform a QUEST-directed 2AFC, choosing which of 2 test stimuli appear most accurate, and curve fitting establishes the width of the stimulus that matches participants' perception of their own body. While this experiment is interesting and topical, connecting with a recent surge of articles on this subject, there are several problems that need to be addressed.

MAJOR ISSUES

The first concerns definitions of the essential concepts for this study. While the title refers to “the internal representation of the body,” and the abstract mentions “the implicit representation of the body”, the meaning of these phrases is never defined. On pg 6, line 18, the authors make it clear that it's not the same as the body image (which is also not defined, even though this phrase has been used to mean various different things in the literature – please fix this), but don't explain in what way the two are different. It doesn't seem to refer to the current perceptual “readout” either (e.g. the “online” representation of the stimulus being viewed, such as the adaptation body or the test stimulus). (Please also explain what is meant by “central effects” – pg 6 line 14.). Without these definitions, it is difficult to fully comprehend what the thrust of the paper is, but it seems that it may be similar to recent work by Ambroziak et al., who distinguished the possible effect of adaptation on the perception of the test image vs. the effects on “internal body image”. Clearly the authors have read this study, yet it is only given a token citation, saying “Ambroziak et al. (2019) used avatar images presented on a small computer screen”. Their approach and findings are not covered. This is surprising, given that the two papers seem to address at least similar issues.

Response 11

We have expanded the introduction to make our motivation clearer and to define our terms and concepts more clearly. Ambroziak et al. did not use realistically sized pictures of their own participants in their experiments and their psychophysical technique for identifying the participants' size estimates was subtly but importantly different from ours. Their technique honed in on the image that ACTUALLY matched their body representation whereas our honed in on an image that is equally like their body image as a reference image. Therefore, our participants never actually saw an image that matched their representation. Instead, they were forced to rely on their own internal representation. The result of our process was two images that were “equally similar to their internal representation”. We have emphasised this point more in the revised ms. Ambroziak et al. were also not interested in the time course of the decline in adaptation effects. However, the results of Ambrosiak are broadly similar to our own: after exposure to a thin image participants judged both themselves and others as thinner and *visa versa*. We have now referenced their findings more explicitly. Interestingly, they found that the perception of other bodies was also affected by their adaptation procedure. However, since they

did not adapt their participants to images of their own bodies but only to a stylized avatar the transfer of effects from “self” to “average” or to famous individuals like Kate Middleton cannot be assigned directly to our results. Here, our main contribution is the time course of decline of the adaptation effects and how they vary with body satisfaction and gender. Ambrosiak et al. explicitly prolonged their adaptation effects by the addition of top-up exposures (since they were asking a different question).

Another recent study to discuss the effects of adaptation on different body representations is Brooks et al. (2021). This article also relates to the description of the pattern of results, which is at odds with most adaptation papers. In the current study, the authors write: “Generally, after participants view a thinner picture of themselves, it leads them to judge their perceived body size to also be thinner...”. In fact, the opposite is true (see Hummel et al, 2012; Brooks et al., 2016 already cited, plus Bould et al., 2018 & 2020 not yet cited). The description of the results as overestimation or underestimation is also discussed in Brooks et al (2021) and relates directly to the issue of what kind of representation is being affected by the process of adaptation. This problem affects many parts of the manuscript, including the description of results and discussion, e.g. on pg 17 lines 16-22. The results of previous body adaptation experiments show repulsion as described here (and as in low-level adaptation experiments, plus face adaptation results), and I believe that the current results may be in line with these (although given the problems with the methods, I can’t be sure). If the results of the current study are out of step with previous body adaptation studies, then an explanation for this discrepancy needs to be given.

Response 12

In the Brooks et al. (2019) women were exposed to multiple other women who were either larger or smaller than them which led to a change in the participants’ perception of the norm and, in contrast, to themselves feeling different from that norm. After being surrounded by fat women, their perception of the norm became fatter, and they consequently felt themselves as thinner than this readjusted “norm”. This is consistent with Rhodes et al. 2013 argument concerning comparing oneself to the norm.

Bould et al. 2020, Hummel et al. 2012, 2013, Ambrosiak et al., Brooks et al. 2012, and the present study all found people judged their own bodies as smaller after viewing smaller bodies in contrast to the Bould et al. 2018 study. To explain the discrepancy in their two results, Bould et al. 2020 proposed that somehow their participant’s perception of the avatar’s used as their assessment tool may have become distorted. Brooks et al. 2016, consistent with Bould et al. 2018, found aftereffects from exposure to distorted bodies in the opposite direction (exposure to fatter bodies made people adjust an image representing themselves as thinner) but curiously interpreted this as both them and the image they were adjusting as being distorted in the same direction (fatter after exposure to a fatter image). The useful and insightful Brooks et al. 2021 review came out after our paper was submitted. It has helpful in expanding our discussion and reference to it has now been added as it addresses these contradictions head on.

We have expanded our discussion to take include these studies and a more explicit statement of where we sit on this issue.

The second major issue concerns the methodology, the details and rationale of which needs to be

explained in full. There is insufficient detail on the procedure and data analysis. For example, we are told “Participants identified which interval contained the image that most closely matched their perception of their own body”, yet the Data Analysis section states: “To visualize and confirm the QUESTs performance, the participant’s decision (0 for smaller, 1 for bigger) was plotted against the distortion used for each trial...”. It is not clear how responses of “distorted image”/“undistorted image” are transformed into “smaller”/“larger”. Further, it is not clear why a logistic function is used for curve-fitting when QUEST’s underlying assumption is that performance reflects an underlying Weibull function. In any case, an example psychometric function fit might help readers to better understand this stage of the experiment.

Response 13

We have explained our method more in the introduction and clarified the methods section further. The logistic was used as a method that allows clear visualization of the data. The logistic was fit using the decisions used by the QUEST to make the image wider or narrower. For most participants the two methods of obtaining the end point were extremely close. Using both methods allowed us to more easily identify outliers. We like your idea of adding a figure to make this clearer and have added a new Figure 4.

Further, the rationale of the method is unclear. The authors state: “In order to assess this, we used the novel psychophysical method described in our previous studies (D’Amour & Harris, 2017, 2019, 2020). This method, since it does not involve participants ever actually seeing the image-size that they judge as their own, gives us a direct estimate of their body representation as opposed to their body image which is vulnerable to many cognitive factors.” Given that the QUEST procedure involves the presentation of many different sized bodies, it seems highly likely that each participant will see the image size that they judge as their own on several occasions. Even if they did not see a body stimulus with this width, why would this mean that they’re judging the “body representation” (again – this needs to be very clearly defined)? The authors need to spell this out.

Response 14

We have explained the method in much more detail now both in the introduction and in the methods. It is true they may transiently have seen the image that matched their body representation (see the new Figure 4 that shows that approximately happening sometimes) but in that case the participant would presumably choose the test image and the QUEST would resize the image away from their choice. We have rephrased our description of the method from “never sees” to “only transiently see” acknowledge this.

Ambroziak et al. (2019) are clear that adaptation affects perception of the test images, not the stored body image. If this is true, then in the current study, adaptation should affect the perception of both of the test images approximately equally, and hence no effect should be shown. For example, after adapting to widened bodies, both test images should look thinner than they really are. This would predict that there should be no measured effects of adaptation, unless the adaptation affects the appearance of one image more than the other. It makes me wonder what the reported effects really represent.

Response 15

If, as Ambroziak et al. claim, the perception of the image(s) is affected by the adaptation procedure, it is true that both the test and reference image would be misperceived. Unfortunately, this would not cancel out any effects of adaptation (which would have allowed us to address their hypothesis directly) but it could cause a displacement on the vertical axis of Figure 4.

It is also interesting that this method is described as “novel” when it appears very similar to the method used by Sekunova et al. (2013) but I can’t be certain due to the lack of details. This paper should be cited.

Response 16

Reference to Sekunova et al. (2013) has now been added. They did an interesting experiment to see if their adaptation effects (to a distorted, headless, 15cm manikin) were robust to changes in pose and viewpoint of a test manikin. They used a method of constant stimuli and were asked to judge which of two examples simultaneously shown 300ms later, a stimulus “body” was more like. This method is quite different from our use of an adaptive staircase using live-size photographs of the participant themselves. Ours required participants to use their own idea of their own body size to make the judgements and our psychophysical method used an adaptive staircase. We hope that our methods have now been clarified sufficiently to make these important differences clear.

I would also recommend greater consistency of phrasing throughout the manuscript. Occasionally, adaptation is referred to instead as “exposure” and sometimes as “feedback”. The latter example is particularly confusing, as it does not seem to be feedback in the conventional sense, unless the participant is being (deceptively) told “this is your actual body size”, which I don’t believe they are (please confirm). Adaptors should either be described as bigger/smaller (as in figures) or wider/narrower, but not both.

Response 17

We have carefully gone through the ms and figures to make sure all such references are consistent, using the terms “wider” and “narrower” as most accurately reflecting our procedure.

MINOR ISSUES

Pg 3, line 21 onwards. See also Zopf et al. 2021, who showed that visual adaptation caused an aftereffect of body size but did not affect perceived tactile distance.

Response 18

This paper is still in an “ahead of print” version, but reference added.

Pg 6 line 1. “factor” should be “factors”

Response 19

Corrected

Pg 6 Line 10. There was no adaptation in the Urdapilleta et al., 2010 study.

Response 20
Deleted

Pg 5 line 12 “How much of this is a simple visual after effect, comparable to low-level adaptation...” This statement is strange, given that pg 18 cover the evidence that body adaptation is high level. Also, see Pg 17, line 23

Response 21
Deleted.

Pg 6, line 7 “We also expected that there would be differences between the adaptation directions – such that adapting to a wider body would cause greater changes in accuracy than adapting to a narrower body.” Why? This hypothesis needs to be justified.

Response 22.
Justification has been provided, see also response 3.

Pg9 line 9 2.2.4. Altering visual feedback with images – This section seems to be procedure, not Materials/Stimuli

Response 23
Moved.

Pg 9 Figure 2. Here, the height of the images seems to vary between the 3 images. Please explain.

Response 24
The height did not vary. Corrected. And the figure relabeled to be consistent with the text.

Pg 11, line 5. “separated by a blank white screen.” For how long?

Response 25
Time of blank screen (1.5s) added.

Pg 15 Figure 4. Remove title from graph. Legend should match text. Don't plot a point at zero,

Response 26
Title removed. Legend adjusted to match insert and to be consistent with the rest of the paper (see response 17). There is no point at zero as all the graphs start at 0 by definition.

Pg 16 line 8. These findings clearly answer the question of whether visually manipulating body representation can alter accuracy values...” I would disagree with this. The idea that

representation has been manipulated is inferred, given the effects measured. It is not a given, whose effects are then assessed.

Response 27

We have toned down this statement.

Pg 16 line 13” We interpret these effects as altering the body representation as opposed to being a simple perceptual effect that resulted from fatiguing particular units or channels.” This comment seems outdated. It has been a very long time since the theory that adaptation is due to fatigue was popular or considered credible. More recent accounts suggest recalibration. A reference more recent than Blakemore 1970 should probably be given.

Response 28

We have removed speculation as to the origin of short-lived adaptation illusions and agree there are probably both fatigue and recalibration aspects.

Pg 16, line 23 “If our adaptation effects were due only to low-level visual perception adaptation mechanisms then everyone should be similar and show comparable levels of adaptation.” Again, it is clear that body adaptation is high level. But even so, the authors need to justify this statement. Why should “everyone be similar”?

Response 29

The logic is that perceptual effects (such as aftereffects relying on low-level sensory changes) should depend only on sensory history. However, body representation, being high level, is subject to more personal factors and is thus more likely to show individual differences – some of which are illustrated in figure 5. The sentence has been rewritten to make this logic clearer.

Pg 18 line 7. “The body representation’s susceptibility to change after such brief experience suggests body representation may be even more plastic than previously thought.” Given that there have been approximately 20 studies to date that demonstrate body size adaptation, should we really be surprised?

Response 30

This statement has been much toned down to point out that our findings add to an emerging literature.

Response to the reviewers

Associate Editor Comments to Author (Dr Isabelle Mareschal):

Both reviewers are not convinced that the effect is one of adaptation of the internal representation rather than perceptual adaptation. There are also still confusions about the method itself, notably about whether participants view an image of their internal representation with the distorted image or not (I agree with both reviewers that this seems to be the case although the authors say it isn't). Unfortunately it is not possible to work out exactly what was done from the method section. I don't want to have a protracted back and forth with the reviewers so please ensure you answer all questions clearly in your reply and make changes in the manuscript. Notably the description of the procedure (e.g. is the "reference" different from the representation?) and the corresponding evidence for adaptation of body representation per se.

RESPONSE: Thanks for the opportunity to clarify our method. See detailed responses below.

Reviewer comments to Author:

Reviewer: 1

Comments to the Author(s)

Thank you to the authors for revising the manuscript. The added details in the Method were helpful for understanding the direction of the effects.

This new sentence in the Discussion sums up the explanation of the results that seems most convincing to me: "...after exposure to a wider body, the test photographs could have appeared narrower, meaning a wider one would need to have been chosen to match their own body representation", i.e., a repulsive visual aftereffect. This contrasts with the author's preferred explanation, which seems to be that adaptation is not affecting perception of the images shown in each trial but is rather affecting an internal sense of the body that is independent of these.

This is relevant to the previous paragraph in the Discussion, where the direction of the effect is cited as a reason to think that it is not a visual aftereffect. Because, actually, the direction of the effect *is* consistent with a typical repulsive visual aftereffect. Also, the sentence "Our technique of not asking our participants to view their own image makes this less likely" doesn't seem relevant, because one would expect a visual aftereffect to act on the perceived size of an image regardless of whether that size matches the participant's own body size or not.

RESPONSE: We concur that the direction of our effect is in the same direction as a repulsive visual after effect but maintain that this is the only similarity. Hummel et al., have already considered this issue and confirmed that the adaptation effect of looking at distorted bodies is specific to bodies and does not transfer to geometric shapes as a regular visual after effect would be expected to do. If our results could be explained as "repulsive visual after effect" then we would expect everything in the image to be scaled up (or down depending on the adapting stimulus) and the slopes of the psychometric functions to be correspondingly affected also (adapt wider, slopes shallower; adapt narrow, slopes steeper). In order to further support our position, we have now included the slopes of the psychometric functions in a revised version of Figure 5

which shows no such “global scaling” effect. The fact that our results were somewhat variable as a function of gender and body dissatisfaction also argues against purely visual aftereffects

We have revised the text of the discussion to be consistent with the direction of the effect.

– A minor thing I found slightly confusing is that it’s stated a couple of times that with the images shown in each trial “neither ... matches their actual idea of themselves”, but actually the reference image shown in each trial is an undistorted image that matches pretty closely with the viewer’s sense of their own body size at baseline (though less so after adaptation).

RESPONSE: It is true that an “undistorted image” is always one of the choices, but this does not necessarily correspond to the person’s internal representation. And, as can be seen from Figure 4 it is not the image that is regularly chosen. The text has been further revised to make our method clearer.

Reviewer: 2

Comments to the Author(s)

Thank you for the opportunity to take a look at the authors’ responses to my first review. Although I acknowledge that they have amended various parts of the manuscript, I regret to inform that I am no closer to recommending publication than I was after the first review. The major concerns at the time of the first review concerned the definition of “representation”; the details of the methodology, and the assertion that this method “gives us a direct estimate of their body representation”. The first and second problems interfered with my ability to fully critique the latter. The methods used for this study are now better explained, and the essential concepts for this study (the “internal representation of the body”) are better defined (“we consider it a kind of memory”). However, now that I understand the concepts and methods, I have more serious concerns with the assertion that the method measures changes in the representation of your own body in memory.

In the revised document, it is again stated: “Thus, this method gives us a best estimate of their body representation as ultimately in the procedure they are forced to rely on accessing their internal body representation to decide which of two photographs neither of which actually matches this representation is closest to it.”

In the reply to reviewer #1, it is stated: “Our psychophysical task is unique because the subject is never shown the actual image that matches their own. Instead, they need to compare their internal idea of their own body size with the presented image. We then end up with two images, neither of which actually matches their representation but from which the size of their representation can be induced. We are not able to say at which level these changes occur – only that they cause the internal representation to be altered. That is the main contribution of this paper – which we emphasize in the title.”

Firstly, the authors now acknowledge that the participants in fact do see an actual image that matches their body, but “only transiently”. But my re-reading of the (now clearer) methods

section makes it clear that they in fact do see an image that matches their body size on every single trial. From section 2.3: “Each trial consisted of two 1.5s intervals – one interval containing an undistorted natural-size image of themselves (the reference image) and one interval containing a distorted image.”

RESPONSE: It is indeed the case that the undistorted image is always one of the choices, but this does not necessarily correspond to the person’s internal representation. Indeed, as Figure 4 shows, they regularly do not choose the undistorted image as being the image that “most closely matches your image of your own body” and end up at the point where the chosen image and the undistorted image are chosen as equally matching the person’s representation of themselves with neither image (including the undistorted image) ACTUALLY matching it.

Apart from that, even if the participants hadn’t seen such an image, it is not clear why this should somehow inoculate them from experiencing any kind of conventional high-level perceptual adaptation that affects the appearance of the visual stimuli shown on screen.

It is clear that any judgement in this manuscript must involve the comparison of the visual stimuli (in this case both the reference and test image) with the mental representation of the participant’s own body (see Brooks et al., 2021).

RESPONSE: We agree completely, and hope that our edits make this clear.

It is of course possible that biased results could arise from changes in the perceived size of the visual images on screen, or a change in the size of the mental representation. Just as in low-level aftereffects, or in high level aftereffects such as those that have been shown to affect faces, visual adaptation is assumed to affect the former. Why the authors assert that their adaptation instead affects the latter is unclear.

RESPONSE: We assert that our effects are operating at the level of the body representation because (a) we do not experience a general increase in scaling as would be expected from a low-level after effect - as shown by the slopes of our psychometric functions not changing and (b) the time course of the effect (over 18 mins) is much longer than would be expected from a low-level adaption after only 5 mins of exposure with no “top ups”.

In the revised discussion, the authors now acknowledge that they “appreciate that this could be a consequence of the photographs themselves becoming perceptually distorted”. Why this is an afterthought, and not the primary assumption is not clear. Regardless, given the acknowledged possibility that the results might not involve any distortion of the body representation in memory, the authors must not make statements to the effect that the body representation has changed as if it were a fact throughout the document (e.g., title, abstract, conclusions, etc.)

RESPONSE: See response to reviewer 1 above: We concur that the direction of our effect is in the same direction as a repulsive visual after effect but maintain that this is the only similarity. Hummel et al., have already considered this issue and confirmed that the adaptation effect of looking at distorted bodies is specific to bodies and does not transfer to geometric shapes as a regular visual after effect would be expected to do. If our results could be explained as “repulsive

visual after effect” then we would expect everything in the image to be scaled up and the slopes of the psychometric functions to be correspondingly affected also (adapt wider, slopes shallower; adapt narrow, slopes steeper). In order to further support our position, we have now included the slopes of the psychometric functions in a revised version of Figure 5 which shows no such “global scaling” effect. The fact that our results were somewhat variable as a function of gender and body dissatisfaction also argues against low-level retinotopic visual aftereffects

We have revised the text of the discussion to be consistent with the direction of the effect.

A new section in the discussion reads: “A recent review by Brooks et al. (2020) reviewed several key points surrounding adaptation in body size and provided support for high-level adaptation of body representation.” That is not my reading of Brooks et al., 2020. Their discussion of these studies is very much in the tradition of visual adaptation (low and high level) with the core assumption that it affects the current perceptual readout, not the stored representation of the body (see also Brooks et al, 2021). Also, in this section, the authors conflate the issue of low-level vs high level adaptation with the issue of whether exposure to body stimuli affect the perceptual readout or the stored representation of their own body. These two are entirely separate issues, yet this confusion is found in several places in the manuscript. As far as I can see, there is no doubt that low-level (i.e. retinotopic) adaptation is not responsible for body size aftereffects, but to say that this means that it must affect the stored representation does not flow logically. Further, the fact that the results were somewhat variable as a function of gender and body dissatisfaction may argue against low-level retinotopic visual aftereffects, but it does not argue against high level visual aftereffects. Note that attention is well known to moderate body aftereffect size. Further, Stephen et al., has previously shown the effect of attention on body size and shape aftereffects, as moderated by body dissatisfaction.

RESPONSE: I think we are agreed that the changes we (and others see) are not the result of low-level adaptation. The question is, are these effects (as shown here and by Hummel et al.) at the level of the stored representation (as we claim) or some other form of long-lasting high-level body-specific effect such as you seem to be suggesting. The transient effects of attention on perception are indeed unlikely to operate at the level of body representation and are probably not specific to bodies. Any attention affects here would be expected to be in the same direction after adapting to either “wide” or “narrow” under our present protocol.

So, the question becomes: can there be high-level adaptation effects that specifically alter perceived body size but that do not involve the internal representation of the body? Brooks et al. 2020 state that “So long as body-size and -shape aftereffects persist at least until the adapted individual next sees his or her own figure, it is possible that this distorted body percept may be “stored” when integrated into the observer’s internal body representation” (pg 138). Hence our reading of the paper as cited in our text (and this quote is referred to again in Brooks et al., 2021). They conclude “neural substrates have been shown to be high-level, non-retinotopic structures whose neurons show some selectivity for identity (e.g., self vs. other) and gender” but without specifying on whether this corresponds to an internal representation of the body.

It would be interesting to look for non-visual effects after visual adaptation in the vein of Zopf et al. (2021) although it maybe that, at least in sighted individuals, the representation is primarily visual.

However, all is not lost. Although the current report of an attractive aftereffect caused by a change on the stored representation is controversial and out of step with previous research, when viewed as a distortion of the current visual percept (i.e., of the reference and test stimulus) the results make sense perfect. In this perspective, the aftereffects shown here are negative (i.e., the visual stimuli look distorted in the opposite direction to the adaptors), as they are in all other body aftereffect studies that I am aware of. (Note the new text: “after exposure to a wider body, the test photographs could have appeared narrower, meaning a wider one would need to have been chosen to match their own body representation.”) In this case, the current study is a neat demonstration that the conventional aftereffects of body size can be produced from life-size stimuli (never been shown before) and can last up to 18minutes (also a novel finding), with some variation associated with gender and body dissatisfaction (this is not so novel, but is interesting nevertheless). Why not drop the controversial claims about “body representation” and simply write up the results to emphasize these useful observations? Although this would involve substantial changes to the manuscript, I believe that such a manuscript would make a valuable contribution to the literature.

RESPONSE: We appreciate your supportive comments pointing out some of the unique features of our study, namely that using actual size images and, critically, that our adaption effects have a time course in excess of 18 mins. To date Cazzato et al. (ref added) appears to be the only attempt to assess the duration of the effects of exposure although they were not actually measuring perceived body size.

We hope that we have been able to convince you that our original claims stand and are consistent with the work of the Macquarie University team.

Appendix C

Response to reviewers:

Associate Editor Comments to Author (Dr Isabelle Mareschal):

Reviewer 2 remains unconvinced that your result is solely attributable to adaptation of the internal representation. Although altering the internal representation may play a role, it is not clear to me that this is the only explanation and/or that other factors may not also be involved. However, the paper does present an interesting method and results, and it has been through a few rounds of review now, so in the interest of time, I will make a final decision on the paper. I will not send it back to review but I need you to provide a more nuanced framing of the interpretation / discussion that leaves room for a role / influence of other (higher level) factors. If you can do this please resubmit your paper with a clear description what you changed and where for me to evaluate.

RESPONSE: We thank you for this decision and have included a discussion of the other factors that may play a role in creating our results. It seems that the reviewers agree with us that we are not looking at low-level effects and we have now explicitly mentioned some alternatives and removed reference to ‘internal representation’ in the title.

Reviewer comments to Author:

Reviewer: 1

Comments to the Author(s)

Thank you to the authors for considering my comments.

It still seems to me that, following adaptation, if the participant’s internal representation of their own body size is unaffected, but their perception of the body width in the images presented on-screen is influenced by adaptation, they would show the same results to those reported here. (e.g., following adaptation to wide images, participants would select a wider test photo as being more similar to their internal representation, because they would see that test photo as being narrower than it really is).

The point about ‘not asking participants to view their own image’ doesn’t seem relevant to this. The arguments against it being a low-level visual aftereffect don’t seem to preclude it from being a high-level visual aftereffect. The results appear equally consistent with a body-shape-specific, repulsive effect of adaptation on the appearance of seen bodies, rather than the attractive effect on the internal representation of the body that the authors describe.

RESPONSE: We have acknowledged these alternative explanations for our results.

I didn’t follow why the slopes of the psychometric function would be diagnostic about whether the results reflect a visual aftereffect or not. Perhaps the logic could be spelt out more here.

RESPONSE: The logic is that if our effects were caused by everything appearing for example, narrower (and therefore needing to be made larger to compensate) then such a global effect would also have shown in the slopes of the psychometric functions (adapt

wider, slopes shallower; adapt narrow, slopes steeper) as the functions were stretched or compressed.

Reviewer: 2

Comments to the Author(s)

I have now had time to peruse the second set of revisions made by the authors, and the responses to reviewers.

Overall, the authors have chosen to double down on their original assertions, principally that “this method gives us a best estimate of their body representation as ultimately in the procedure they are forced to rely on accessing their internal body representation” and that “visual feedback impacts the internal representation of body size”. I remain unconvinced.

In my previous review, I noted: “it is not clear why this [the still-disputed claim that the participants never see an image that matches their memory representation of their own body] should somehow inoculate them from experiencing any kind of conventional high-level perceptual adaptation that affects the appearance of the visual stimuli shown on screen.” For this, the authors had no answer (unless it is the words “We agree completely, and hope that our edits make this clear” although to be fair, I doubt that this was intended to be a direct response to this particular concern).

Response: We have clarified this point still further.

Repeatedly, in responses to reviewers, and in the revised manuscript, the authors try to convince us that the effects they show are unlikely to be due to low-level (i.e., retinotopic) adaptation (note that in the revision the phrase “low-level” has been added in 3 locations). Further, new data on psychometric function slopes are provided. The authors claim: “If our results could be explained as “repulsive visual after effect” then we would expect everything in the image to be scaled up and the slopes of the psychometric functions to be correspondingly affected also (adapt wider, slopes shallower; adapt narrow, slopes steeper).” Unfortunately, the lack of statistically significant differences between slopes for each condition is unconvincing for several reasons. First, the QUEST procedure used here may be effective for determining a psychometric function mid-point, but it is far from optimal for determining slope. Some function parameters are calculated from as few as 16 trials (see figure 4). Many more trials, targeted at upper and lower points of the curve, would be required to achieve any kind of reliable slope measurement. On top of this, null results (especially those produced by noisy measurements) are never convincing.

Response: We acknowledge that the psychometric design of the experiment was not optimized to look at the slopes. However, all curves were fitted with at least 50 points (25 from each of two interleaved staircases. Figure 4 shows just the fit to one staircase for clarity.

If our effects were caused by everything appearing for example, narrower (and therefore needing to be made larger to compensate) then such a global effect would also have shown in the slopes of the psychometric functions (adapt wider, slopes shallower; adapt narrow, slopes steeper) as the functions were stretched or compressed.

That low-level adaptation is not responsible for the effects here is not disputed by anyone.

Response: Excellent, and we have now cemented that argument further.

That body adaptation is a high-level effect has been shown by several studies (including Hummel et al., of which the authors are aware). Yet as I stated previously, this does not leave effects on memory as the only logical alternative. High-level perceptual (not memory) aftereffects have been widely reported for over 20 years, initially in the face perception literature, and are the most obvious explanation here. In my previous review I suggested that the authors report their interesting effects in this more conventional framework, but sadly they have declined this opportunity. To assert that the effects shown here are due to effects on stored representations in memory would require additional experiments. Without clear evidence, such claims are speculative at best. As in my second review, I maintain that the assumption that memory must be affected must not form the framework for the entire paper, and certainly cannot appear in the title, abstract or conclusions.

Response: We have included further discussion of the other factors that may play a role and have expanded on the other possible frameworks that may influence our findings and interpretations. We have also changed the title and made small revisions describing some alternative factors that can play a role.

I appreciate the hard work that has gone into this paper, but given that the authors' have declined to take my previous two sets of comments on board, I regret that I am unable to recommend publication of this manuscript.